# *Opuntia* spp. in Human Health: A Comprehensive Summary on Its Pharmacological, Therapeutic and Preventive Properties. Part 2

**DOI:** 10.3390/plants11182333

**Published:** 2022-09-06

**Authors:** Eduardo Madrigal-Santillán, Jacqueline Portillo-Reyes, Eduardo Madrigal-Bujaidar, Manuel Sánchez-Gutiérrez, Jeannett A. Izquierdo-Vega, Julieta Izquierdo-Vega, Luis Delgado-Olivares, Nancy Vargas-Mendoza, Isela Álvarez-González, Ángel Morales-González, José A. Morales-González

**Affiliations:** 1Escuela Superior de Medicina, Instituto Politécnico Nacional, “Unidad Casco de Santo Tomas”, Ciudad de México 11340, Mexico; 2Escuela Nacional de Ciencias Biológicas, Instituto Politécnico Nacional, “Unidad Profesional A. López Mateos”, Ciudad de México 07738, Mexico; 3Instituto de Ciencias de la Salud, Universidad Autónoma del Estado de Hidalgo, Ex-Hacienda de la Concepción, Tilcuautla, Pachuca de Soto 42080, Mexico; 4Escuela Superior de Cómputo, Instituto Politécnico Nacional, “Unidad Profesional A. López Mateos”, Ciudad de México 07738, Mexico

**Keywords:** *Opuntia* spp., neuroprotective, antiulcerative, antimicrobial, antiviral, skin wounds, anti-inflammatory effect

## Abstract

Plants of the genus *Opuntia* spp are widely distributed in Africa, Asia, Australia and America. Specifically, Mexico has the largest number of wild species; mainly *O. streptacantha*, *O. hyptiacantha*, *O. albicarpa*, *O. megacantha* and *O. ficus-indica*. The latter being the most cultivated and domesticated species. Its main bioactive compounds include pigments (carotenoids, betalains and betacyanins), vitamins, flavonoids (isorhamnetin, kaempferol, quercetin) and phenolic compounds. Together, they favor the different plant parts and are considered phytochemically important and associated with control, progression and prevention of some chronic and infectious diseases. Part 1 collected information on its preventive actions against atherosclerotic cardiovascular diseases, diabetes and obesity, hepatoprotection, effects on human infertility and chemopreventive capacity. Now, this second review (Part 2), compiles the data from published research (in vitro, in vivo, and clinical studies) on its neuroprotective, anti-inflammatory, antiulcerative, antimicrobial, antiviral potential and in the treatment of skin wounds. The aim of both reviews is to provide scientific evidences of its beneficial properties and to encourage health professionals and researchers to expand studies on the pharmacological and therapeutic effects of *Opuntia* spp.

## 1. Introduction

The Traditional Medicine/Complementary and Alternative Medicine (TCAM) concept includes any practice, knowledge and belief in health that incorporates medicine based on plants, animals and/or minerals, spiritual therapies, manual techniques and exercises applied individually or in combination to improve human health. The World Health Organization (WHO) considers that TCAM have shown favorable factors that contribute to an increasing acceptance worldwide, such as easy access, diversity, relatively low cost and, most importantly, relatively low adverse toxic effects in comparison with allopathic medicine where these effects are frequently attributed to synthetic drugs. For this reason, TCAM continues to be used by different populations to treat and/or prevent the onset and progression of chronic and infectious diseases [1,2,3].

Throughout human history, plants and their phytochemicals have played an important role at improving human health care. *Opuntia* species have specifically shown many beneficial properties and high biotechnological capacity. These plants classified as angiosperm dicotyledonous are the most abundant of the Cactaceae family and are importantly distributed in America, Africa, Asia, Australia, and in the central Mediterranean area. Due to their capacity to store water in one or more of their organs, they are considered succulent plants whose cultivation is ideal in arid areas since they are very efficient to generate biomass in water scarcity conditions [4,5,6,7].

Most opuntioid cacti have flat and edible stems called cladodes (CLDs), paddles, nopales or stalks. Generally, young CLDs (also called nopalitos) are eaten as a vegetable in salads, while their fruits (called cactus pear fruits, tunas or prickly pear fruits (PPFs)) are widely eaten as fresh seasonal fruit. PPFs are oval berries with lots of seeds throughout all the pulp and a semi-hard bark that contains thorns. They are grouped in different colors (red, purple, orange/yellow, and white). Generally, the fruit with white flesh and green skin is the most consumed as food [4,5,6,7]. Some evidences indicate that *Opuntia* plants have been consumed by humans for more than 8000 years and due to their easy adaptation and spread in different types of soil, their domestication process has favored the constant collection of CLDs and PPFs by man [7,8,9,10].

## 2. Impact of the *Opuntia* Genus in Mexico and Other Countries

The Cactaceae family includes about 200 genera and 2000 species classified into three to six subfamilies. The Opuntioideae subfamily comprises between 15 and 18 genera, *Opuntia* being the most diverse and widely distributed genus in the American continent [10,11,12]. However, Mexico has the largest number of wild species. The most representaive are *O. streptacantha* (OS), *O. hyptiacantha* (OH), *O. albicarpa* (OA), *O. megacantha* (OM) and *O. ficus-indica* (Figure 1). The latter is highly cultivated and domesticated species due to its nutritional, medicinal, pharmaceutical, and economic impacts. It is believed to be a secondary crop with fewer thorns derived from OM, (a native species from central Mexico) [4,7,8,10,11,12]. Currently, *O. ficus-indica* (OFI) has become as important a vegetable crop as corn and agave-tequila; its economic relevance is significantly increasing in our country and in other parts of the world, especially for improving health when nopal and prickly pear are included in a diet. Therefore, the OFI domestication process has favored changes in the texture, flavor, size, color, quantity and quality of the cladodes and their fruits [4,7,8]. Mexico and Italy are the main producing and consuming countries of the approximately 590,000 ha cultivated around the world. The Annual Mexican production can reach 350,000 tons; for this reason, our country represents approximately 90% of the total production worldwide. In addition, Mexico is the main producer of prickly pear fruits, representing more than 45% of world production; however, only 1.5% of this production is exported, due to various factors such as a low level of technology in its production, climatic changes, the lack of a marketing plan (supply/demand) and that it is a seasonal product (it is obtained between the months of June to November) [4,7,12,13].

In relation to the impact of *Opuntia* species in other countries, there is evidence of positive and negative aspects. In the first case, due to their adaptability, which allows them to grow where no other crop can, their plants and fruits (specifically, PPFs) have become an essential and reliable crop for the diet of the inhabitants (As it happens in Ethiopia). Likewise, healers of the Kani tribes in the Tirunelveli hills of the Western Ghats of India, consider the opuntia species spiritual and esential for their first aid remedies; especially *O. dillenii* which is used to treat cough, headache, poisonous animal bites, cold and fever control [14,15].

Unfortunately, because they can survive in arid/semi-arid environments, high temperatures, little rainfall and limited nutrient supply, they are considered an invasive alien species in some South African and Kenyan communities that can threaten biodiversity and food security. For this reason, *Opuntia* (especially *O. stricta*) is included in the diet of animals (forage for livestock), as a measure to control its spread, mainly in dry seasons [16,17].

In general, in Mexico and other countries, *Opuntia* is a resource that has high agrotechnological potential, both as a food crop and as a base element to obtain derived products, which are used in the food industry (human and animal), medicine, agricultural industry and cosmetology (Table 1).

## 3. Nutritionalcomposition and Mechanisms of Pharmacological Action of the *Opuntia* Genus

Different methods have documented the nutritional value of *Opuntia* spp. Most of these studies coincide in the differences among the phytochemical composition of their plant parts (fruits, roots, cladodes, flowers, seeds and stems) and the wild and domesticated species. These can be attributed to environmental conditions (climate, humidity), the type of soil that prevails in the cultivation sites, the age of maturity of the cladodes, and the harvest season [5,6,7]. The nutritional composition of the different parts of *Opuntia ficus-indica* (L.) Mill. is summarized in Table 2. In general, opuntioid cacti contain a large amount of water (80 and 95%), carbohydrates (3–7%), proteins (0.5–1%), soluble fiber (1–2%), fatty acids (palmitic, stearic, oleic, vaccenic and linoleic) and minerals (Potassium (K), calcium (Ca), phosphorus (P), magnesium (Mg), chrome (Cr) and sodium (Na)). They also have viscous and/or mucilaginous materials (made up of D-glucose, D-galactose, L-arabinose, D-xylose and polymers such as β-D-galacturonic acid linked to (1–4) and L-rhamnose residues linked with R (1–2)) whose function is to absorb and regulate the amount of cellular water in dry seasons [5,6,7,19]. Among the main bioactive compounds of prickly pear highlight the pigments (carotenoids, betalains, betaxanthins and betacyanins), vitamins (B1, B6, E, A, and C), flavonoids (isorhamnetin, kaempferol, quercetin, nicotiflorine, dihydroquercetin, penduletin, lutein), rutin, aromadendrine, myricetin vitexin, flavonones and flavanonols) and phenolic compounds (ferulic acid, feruloyl-sucrose and synapoyl-diglycoside) [5,6,7,19,20,21,22].

Specifically, the cladodes and prickly pears fruits of OFI have shown several kinds of bioactive compounds, among which flavonoids (such as quercetin, kaempferol, isorhamnetin), essential amino acids (Glutamine, arginine, leucine, isoleucine, lysine, valine and phenylalanine), vitamins (B1, B6, E, A, and C), minerals (mainly K and Ca), and betalains [such as betaxanthins (betanin and indicaxanthin) and betacyanins (betanidin, isobetanin, isobetanidine, and neobetanin) [5,6,7,19,20,21,22].

As mentioned in part 1 [23], various studies have shown the action of phytochemicals as substrates to activate different biochemical reactions that provide important health benefits. For that reason, they could be included in the definition of nutraceutical: “Any non-toxic food extract supplement that has been scientifically proven to be beneficial to health both intreating and preventing diseases” [5,23,24]. Different authors agree that the carotenoids are important compounds with great benefits for human health, related to the prevention and reduction of the development of some diseases, such as cardiovascular diseases, cancer and macular degeneration and that taurine (semi-essential amino acid) is involved in the modulation of the inflammatory response with potential antioxidant. As well as, that some plant sterols are incorporated into foods intended for human consumption to lower blood cholesterol levels [25,26]. On the other hand, the scientific evidence suggests that the phenolic acids (hydroxycinnamic acids and hydroxybenzoic acids), flavonoids, lignins and stilbenes have a high antioxidant potential that has been related in many health benefits such as prevention of inflammation, cardiovascular dysregulation, and neurodegenerative diseases [25,27]. In this same approach, it is known that the flavonoids are a group of bioactive compounds that exhibit many effects in the protection of the body, and their regular consumption is associated with reduced risk of several chronic diseases (especially, for its antioxidant, antiviral and antibacterial capacities). Finally, the Betalains are powerful radical eliminators in chemical systems and has act as efficient antioxidants in several biological models. Potential related as a possible strategy for intestinal inflammation [26,27,28]

In this context, opuntioid cacti reveal different mechanisms of action that can be interrelated and favor their biological effects. In general, they are organized in 7 groups: (I) Inhibition of the absorption of substances, favoring the absorption of protective agents and/or modification of the intestinal flora (action of soluble fiber and ascorbic acid), (II) Scavenging of reactive oxygen species and/or protection of DNA nucleophilic sites (antioxidant action), (III) Anti-inflammatory activity, (IV) Modification of transmembrane transport (effect of short-chain fatty acids and calcium in the diet), (V) Modulation of xenobiotic metabolising enzymes, inhibition of mutagen agents activation and induction of detoxification pathways (flavonoids, polyphenols and índoles), (VI) Enhancement of apoptosis (action of some flavonoids), and (VII) Maintenance of genomic stability (effect of some vitamins, minerals and polyphenols) [3,5,6,7,20,21,22,23,24].

Together, the bioactive compounds of *Opuntia* spp. favor its different plant parts to be considered phytochemically important and associated with the control, progression and prevention of some chronic and infectious diseases. This second review (Part 2) focuses on information from published research (in vitro, in vivo and clinical studies) on its anti-inflammatory, antiulcerative, neuroprotective, antimicrobial, antiviral properties and on the treatment of skin wounds; which will be discussed below.

## 4. Pharmacological, Therapeutic and Preventive Properties

### 4.1. Anti-Inflammator and Antiulcerative Effects

The inflammatory cascade includes a long chain of molecular reactions and cellular processes (highlighting phagocytosis, chemotaxis, mitosis, and cell differentiation) designed as a biological response to different noxious stimuli, including dust particles, chemical substances, physical injuries, bacteria, viruses, and parasites. This cascade is an important factor for the progression of various chronic disorders, such as obesity, arthritis, diabetes, cancer, cardiovascular diseases, eye disorders, autoimmune diseases and inflammatory bowel disorders; therefore, in recent decades it has become a highly studied field of research [29,30]. The stages of inflammation depend on the duration of the process and various immunological factors, classifying them into acute and chronic. The first is characterized by a rapid initial response that can last minutes and/or a few days. There is accumulation of plasma proteins and leukocytes, presenting increased blood flow, swelling, redness, pain and heat. When this response is prolonged, chronicity is established, which leads to an increase in the presence of lymphocytes, macrophages, and mast cells at the site of infection and/or damage. Simultaneously, repair mechanisms that can generate fibrosis and overproduction of connective tissue are activated. In general, inflammation is a vital response of the immune system; however, a chronic process can induce secondary consequences in the biological response associated with an increased risk of chronic diseases. This usually occurs through infections that are not cleared by endogenous protective mechanisms or by some type of genetic susceptibility [29,30].

Various investigations have confirmed that inflammation is related to the induction of oxidative stress (OXs) due to the increase in cells (lymphocytes, macrophages, and mast cells) that leads to greater oxygen uptake, increasing the production and release of reactive oxygen species (ROS) in the damaged area. In addition, the activation of signal transduction cascades and alteration in transcription factors (such as nuclear factor kappa B (NF-κB), signal transducer and activator of transcription 3, activator protein-1, factor 2 related to NF-E2, activated T-cell nuclear factor, and hypoxia-inducible factor-1α (HIF1-α)) [29,30]. Likewise, cyclooxygenase-2 (COX-2), nitric oxide synthase (iNOS), expression of inflammatory cytokines (such as tumor necrosis factor alpha (TNF-α)), interleukins (IL-1β, IL-6) and chemokines, are induced. Scientific findings of different anti-inflammatory agents have shown that bioactive extracts and their natural compounds exert their biological properties by blocking signaling pathways, such as NF-κB and mitogen-activated protein kinases (MAPK) [29,30].

On the other hand, Gastric Ulcers (GU) are open sores in the mucosa lining the stomach and/or duodenum. The most frequent symptomatology is pain and burning that can occur between meals or at night, with different durations (minutes and/or days). The worldwide incidence of GU varies depending on age, sex, and geographic location, but it remains a common condition and a major public health problem due to high healthcare costs and life-threatening complications (bleeding, perforation, and obstruction), that favor its high morbidity and mortality [31]. The pathophysiology of GU is multifactorial but is generally associated with the result of an imbalance between the protective and aggressive factors of the gastric mucosae; that is, when there is a significant increase in the acids that help digest food, damaging the walls of the stomach and/or duodenum. Among the harmful factors that favor its incidence are excessive gastric acid and pepsin secretion, increased ROS, *Helicobacter pylori* infection, constant alcohol consumption, and prolonged ingestion of nonsteroidal anti-inflammatory drugs (NSAIDs) [31,32,33]. While gastrointestinal defense mechanisms include mucus secretion, bicarbonate production, nitric oxide (NO), prostaglandin synthesis, normal gastric motility, and adequate tissue microcirculation. Currently, GU treatments are aimed at improving the defenses of the gastric mucosae or counteracting harmful factors. Among the most used are those that reduce gastric acid secretion (H2 receptor antagonists), those that inhibit the proton pump (omeprazole); and antibiotics that control *H. pylori*. However, its high costs and side effects of long-term treatments combined with recurrence of ulcers and some cases of rejection to conventional therapies have motivated the search for new antiulcer agents. In particular, those that improve the quality of ulcer healing to prevent abnormalities in mucosal regeneration and the persistence of chronic inflammation by reducing the infiltration of neutrophils and macrophages (i.e., prevent ulcer recurrence) [31,32,33].

Like anti-inflammatory agents extracted from natural compounds, herbal anti-ulcer medications have also become an excellent source to obtain them. In this sense, *Opuntia* spp. is no exception, and possibly, after the studies related to atherosclerotic cardiovascular diseases, diabetes, obesity and chemopreventive capacity (included in part 1) [23], the anti-inflammatory and antiulcerative evaluation field is of equal relevance to researchers.

Probably, the first study to break into this field of evaluation was aimed to confirm in rodents that a preparation of dried flowers of OFI reduced the discomfort of benign prostatic hypertrophy by suppressing the release of beta-glucuronidase (lysosomal enzyme of the neutrophils) [34]. Subsequently, two research groups, Park et al., (1998) [35] and Loro et al., (1998) [36], continued the studies. In the first, ethanolic extracts of fruits (EEOF) and stems (EEOS) of OFI were analyzed on the acetic acid writhing syndrome and paw edema induced by carrageenan (CRRG) in Sprague Dawley rats. Both extracts decreased writhing and edema; as well as the release of the same lysosomal enzyme. Their results suggested that EEOF and EEOS have analgesic and anti-inflammatory actions, and a possible protective effect against gastric injury [35]. Using similar techniques, the second group of scientists evaluated different lyophilisates (50–400 mg/kg, i.p.) from the fruits of *O. dillenii* Haw (OdHw); the result was similar when the chemical stimuli were dose-dependently inhibited (writhing test) and thermal (hot plate test) in Wistar rats; mainly in doses of 50 and 100 mg/kg [36]. The results of the previous studies motivated the fractionation of a methanolic extract of OFI stems and, together with a model of chronic inflammation induced by adjuvants in mice, β-sitosterol was isolated and identified as a possible anti-inflammatory active ingredient [37].

In relation to the first evidence of the anti-ulcer effect, Galati et al. [38,39,40] found that by previously administering lyophilized CLDs and/or OFI whole fruit juice on experimental ulcers induced by ethanol (EtOH) in rats, a cytoprotective action related to an increase in the production of gastric mucus was exerted. They attributed such effect to the mixture of mucilage and pectin present in OFI [38,39,40]. Table 3 shows the main studies that evidence the anti-inflammatory and antiulcerative effects of *Opuntia* spp. In summary, from 1993 to date, 16 of 39 have been in vitro studies; 20, using laboratory animals (mainly rodents) and 3, developed with patients (clinical studies). Mainly, different types of extracts (methanol (MeOH), hexane (Hx), chloroform (Chl), ethyl acetate (EtOAc), butanol and aqueous) obtained from CLDs and roots of OFI, OdHw and *O. humifusa* (OHF) have been analyzed. In addition, powders from the stems, juices, vinegars and oils of PPFs and/or their seeds extracted from *O. elatio* Mill, *O. macrorhiza* Engelm (OME), OFI and OdHw have been explored. Most studies confirm and agree that the main mechanism of action of both properties (anti-inflammatory and anti-ulcer) is related to its antioxidant capacity, which is attributed to a possible synergistic and/or combined effect between the different bioactive compounds (phenols, flavonoids (such as quercetin, kaempferol, isorhamnetin), betalains (betanin and indicaxanthin), betacyanins, α-pyrones (opuntiol and opuntioside glucoside), pectin and mucilage) present in the chemical composition of the *Opuntia* genus [41,42,43,44,45,46,47,48,49,50,51,52,53,54,55,56,57,58,59,60,61,62,63,64,65,66,67,68,69,70,71,72,73,74,75].

### 4.2. Neuroprotective Effect

Neurodegenerative disease (ND) is a progressive dysfunction and/or loss of neuronal structure and function, generally irreversible, that alters intellectual and cognitive faculties. This disorder occurs in various diseases that affect the central nervous system (CNS) and may be acute or chronic [76,77,78,79]. The first case refers to a condition where neurons are rapidly damaged and can die in response to a sudden insult or traumatic event (such as head injury, stroke, traumatic brain injury, brain hemorrhage, or ischemic brain damage). On the other hand, chronic neurodegeneration is a state where a degenerative process that begins slowly and worsens over time due to multifactorial causes, is experienced; resulting in the progressive and irreversible destruction of specific neuronal populations. Among the most significant chronic neurodegenerative disorders are Alzheimer’s disease (AD), Huntington’s disease, Parkinson’s disease (PD), and amyotrophic lateral sclerosis. As mentioned, the causes of ND are multifactorial and are associated with different types of biological mechanisms, which in general can be summarized as: (a) oxidative stress (EOx), (b) neuroinflammation, (c) excitotoxicity, (d) mitochondrial dysfunction, (e) induction of apoptosis and (f) abnormal protein folding and aggregation [76,77,78,79].

Specifically, imbalanced ROS production and poor antioxidant defense (endogenous and/or exogenous) cause EOx resulting in cell damage, impaired DNA repair system, and mitochondrial dysfunction; accelerating the neurodegenerative process. On the other hand, neuroinflammation involves both the innate and adaptive CNS immune systems, playing an important role in the pathophysiology of DN. Since microglia are the main components of the innate immune defense, and if there are pathological changes within the CNS, they secrete inflammatory mediators (cytokines, chemokines, COX-2), which activate astrocytes to induce a secondary inflammatory response in a population of neurons that respond to a survival process [76,77,78,79].

Excitotoxicity [neuronal death caused by excessive or prolonged activation of glutamate (Glu) receptors by excitatory amino acids or CNS excitotoxins] is also involved in degenerative pathogenesis. Excitotoxins that bind to Glu receptors, as well as pathologically high levels of their release, are known to cause toxicity by allowing a rapid entry of calcium ions (Ca^2+^) into the cell, activating various Ca^2+^-dependent enzymes (iNOS, phospholipases, lipase endonucleases, xanthine oxidase, protein phosphatases, proteases, and protein kinase). These enzymes continue to damage cellular structures (such as components of the cytoskeleton, membrane, and DNA) and/or generate ROS, mitochondrial dysfunction, and other inflammatory responses, which together lead to neuronal death [76,77,78,79].

Finally, combining apoptosis (a highly regulated form of cell death that is triggered by intrinsic and extrinsic signals) and the mitochondria (site of oxidative phosphorylation and cellular respiration), a significant role is played in maintaining a low concentration of Ca^2+^ cytosolic [76,77,78,79]. Excessive uptake of this ion and generation of ROS cause the opening of mitochondrial permeability transition pores, inducing matrix inflammation, mitochondrial uncoupling, and membrane rupture that releases cytochrome-c (Cyt-c) and apoptosis inducing factor. Cyt-c, a caspase-dependent pathway, binds to apoptotic protease-activating factor 1 and procaspase-9 to form an apoptosome complex and activate the caspase-3 pathway, producing apoptotic neuronal death. While the apoptosis-inducing factor, a caspase-independent mechanism, moves to the nucleus and induces the DNA fragmentation, chromatin condensation and subsequently cell collapse [76,77,78,79].

Progressive degeneration and/or neuronal death causes characteristic symptoms such as problems with movement (ataxia) or alterations in cognitive functioning (dementia), and since, unfortunately, most NDs have no cure, conventional treatments focuson improving symptoms and relieving pain. For example, in individuals with PD there are low concentrations of dopamine (DA) in the brain, so the main drugs (Carbidopa-levodopa, Dopamine agonists, Inhibitors of the enzyme monoamine oxidase type B or Inhibitors of catechol -O-methyltransferase), mimic, increase or replace this neurotransmitter (NT) [76,77,78,79]. In the case of AD, cholinesterase inhibitors are prescribed to patients with mild and/or moderate symptoms to prevent the breakdown of Acetylcholine (Ach), a brain neurotransmitter that is related to memory and thought. Unfortunately, these medications often lose their therapeutic effect over time. Another example is the so-called “disease-modifying agent”, “aducanumab”, a human antibody that targets β-amyloid protein (Aβ) to reduce brain lesions (amyloid plaques) associated with AD [76,77,78,79].

Various studies have tried to elucidate mechanisms and possible therapeutic objectives to combat NDs, in order to avoid neuronal damage and preserve the integrity and functionality of these cells; all resultingin a concept known as Neuroprotection. A strategy that includes three approaches: (1) before the onset of the disease to avoid any risk factors might affect the neurons. (2) during the progression of the disease to avoid the spread of the lesion from one neuron to another; and (3) to try to delay and /or stop progressive neurodegeneration [76,77,78,79].

Again, natural products and their bioactive compounds are an excellent source of neuroprotective agents for the treatment of ND. The early agents studied limited their mechanism of action to intervene in NT receptors through agonists and antagonists; the best known example was caffeine, adenosine A2 receptor antagonist, which has been shown to protect dopaminergic neurons in an experimental model of PD induced by 1-methyl-4-phenyl-1,2,3,6-tetrahydropyridine. It is now known that microglial cells also express the A2 receptor, and A2 receptor antagonist or caffeine can reduce the activation of these cells. Therefore, drinking coffee, maintaining a healthy lifestyle and having moderate physical activity has been considered a neuroprotective strategy, and added to the fact that A2 antagonists can protect neurons and minimize the activation of microglial cells, the field of research of phytotherapy continues to be active in the search for new and innovative neuroprotective agents [76,77,78,79].

Since multifactorial pathological mechanisms (EOx, neuroinflammation, excitotoxicity, mitochondrial dysfunction, and apoptosis) are associated with neurodegeneration, current research look for multiplemechanisms of action that intervene in the complexity of the disease with natural neuroprotective agents instead of looking for a single biological objective.

Table 4 shows the main evidence of the neuroprotective effect of *Opuntia* spp. Since the year 2000, when the exploration of this scientific field began, most of the studies have been carried out in vitro (mouse cortical cells, primary cultured rat cortical cells, PC12 cells) where extracts of CLDs and prickly pear fruits have been evaluated. Some phytochemicals (Quercetin, quercetin 3-methyl ether, indicaxanthin, polysaccharides) from 6 species of *Opuntia*; where the most studied are OFI and OFI var. Saboten (OFIS). Basically, neuronal damage and toxicity have been induced by different agents and/or substances; such as xanthine/xanthine oxidase (X/XO), FeCl2/ascorbicFeCl2/ascorbic acid, N-methyl-d-aspartate (NMDA), kainate (KA), oxygen-glucose deprivation (OGD, LPS, AlCl3 and Aβ. Although, probably, the neuroprotective effect of *Opuntia* spp. can be carried out through multiple mechanisms, most authors agree that the antioxidant capacity is the most significant and/or representative [80,81,82,83,84,85,86,87,88,89,90,91,92,93]

### 4.3. Antiviral and Antimicrobial Effects

Despite the incredible progress in human medicine, Viral Infections (VI) continue to be responsible for various chronic and acute diseases. Diseases such as Acquired Immunodeficiency Syndrome (AIDS), hepatitis, and respiratory syndromes, especially the one caused by the severe acute respiratory syndrome virus type-2 (SARS-CoV-2) are associated with high rates of morbidity and mortality [94,95]. Again, natural products are a rich source of bioactive compounds with possible antiviral effects; thus, identifying them is of critical importance. A wide variety of phytochemicals, including coumarins, flavonoids, terpenoids, organosulfur compounds, lignans, polyphenols, saponins, proteins, and peptides, have been found to influence cell functions, membrane permeability, and viral replication. Therefore, natural-based pharmacotherapy may be a good alternative for VI treatment. Antiviral agents can be classified according to their chemical nature or their activity against viral proteins and/or host cellular proteins. Particularly, the antiviral activity can be exerted based on its ability to inhibit any viral entry, viral DNA and RNA synthesis, as well as viral replication/reproduction. Differences in viral structure and replication cycle are crucial to the design of any antiviral medication [94,95].

The antiviral activity of phytochemicals can be established by different biological assays, commonly used to assess cytotoxicity, cytopathic effect, and the ability to block viral cell-to-cell propagation. Purified natural products are considered a rich resource for the development of new antiviral drugs. However, their extraction and isolation can be a difficult process, since many bioactive compounds are present in low concentration in the natural source and due to their complex chemical structures they are not easy to synthesize. In addition, the majorityof the natural compounds are used as unpurified crude extracts, which makes it very important to isolate each biomolecule individually and to establish their pharmacokinetics (absorption, distribution, metabolism, excretion), therapeutic effects, dose and possible toxicity events [94,95].

Among the isolated bioactive compounds with recognized antiviral action is rutin (known as quercetin-3-rutinoside), a flavonoid glycoside effective against avian influenza virus, herpes simplex virus 1 and 2 (HSV-1, HSV-2) and parainfluenza -3 virus [94,95]. Quercetin, an aglycone of rutin, has demonstrated its therapeutic potential against influenza A virus (IFV-A), rhinovirus, dengue virus type 2 (DENV-2), HSV-1, poliovirus type 1 (PV- 1), adenovirus, Epstein-Barr virus, Mayaro virus, Japanese encephalitis virus, Respiratory Syncytial Virus (RSV), and Hepatitis C virus (HCV) [94,95]. Among the mechanisms of action of quercetin are limiting the activity of some thermal shock proteins involved in viral translation (Internal Ribosome Entry Site or IRES) mediated by the non-structural protein 5A (NS5A), inhibiting NS3 protease and viral replication of HCV, reducing endocytosis, blocking viral genome transcription and rhinovirus protein synthesis, and decreasing DENV-2 replication [94,95]. Other flavonoids, such as myricetin (3,3′,4′,5,5′,7-hexahydroxyflavone), quercetagetin (3,3′,4′,5,6,7-hexahydroxyflavone), and Baicalein (5,6,7-trihydroxyflavone) block the reverse transcriptase of the Rauscher murine leukemia virus and the human immunodeficiency virus (HIV). Finally, Baicalin (the glucuronide of baicalein) inhibits the synthesis of DNA and viral proteins of the hepatitis B virus (HBV); it is also active against HIV, DENV, RSV, enterovirus, and Newcastle disease virus [94,95].

Thus far, only three investigations have evaluated the antiviral effect of *Opuntia* spp. In the first, administration of an OS stem extract to mice, horses, and humans inhibited the intracellular replication of several DNA and RNA viruses, such as HSV-2, RSV, HIV, IFV-A, equine herpes and pseudorabies virus. Although a viral inactivation at the extracellular level was also observed, there were no answers about the possible inhibitory components of the extract [96]. On the other hand, Bouslama et al., (2011) analyzed the inhibitory effect of two extracts (aqueous and/or EtOH) from OFI stems on the replication of two enveloped viruses (HSV-2 and IFV-A) and a non-enveloped virus (PV-1). Given that only the EtOH extract showed significant antiviral activity in vitro, two stem chlorophyll derivatives (pheophorbide a and pyropheophorbide a) were isolated; which demonstrated a virucidal effect only on both enveloped viruses. These findings suggest that both phytochemicals could recognize specific glycoproteins of enveloped viruses, preventing their binding to the host cell receptors and inhibiting VI [97]. In the latest study, an antiviral protein (named Opuntin B) from OFI was purified; which shows the total degradationof genomic RNA of the plant and causes a displacement of the electrophoretic mobility of the RNAs of the cucumber mosaic virus (CMV). Using CMV as prey protein and Opuntin B as bait protein, far western dot blot analysis showed no interaction between antiviral protein and viral coat protein [98].

On the other hand, microbial pathogens (MP) can enter a host using different transmission mechanisms, which are generally classified as: (a) direct contact (cutaneous lesions, urogenital tract and/or sexual transmission), (b) indirect contact (contaminated hands and/or inanimate instruments), and (c) airway (inhalation of droplets of different diameters through the respiratory tract and/or ingestion of contaminated food or drink). Regardless of the route of transmission, MPs are responsible of producing various diseases that generate public health problems and cause excessive economic costs [99,100]. Unfortunately, antimicrobial resistance has also become an increasingly important and pressing global problem, as of the millions of people who acquire bacterial infections, approximately 70% of cases involve strains that are resistant at least to one drug [99,100,101]. Therefore, in response to this problem, pharmaceutical companies are focusing their efforts on improving antimicrobial agents; however, the researchers acknowledge that they are reaching the end game in terms of alterations to their chemical structures. For this reason, natural products can be a rich source of anti-infective agents that work at different target sites and can replace synthetic compounds [99,100,101].

Obtaining natural antimicrobial agents also has an impact on food preservation to prevent disease transmission after ingestion of contaminated food and/or beverages. These bioconservatives must keep and preserve nutritional values and/or guarantee food safety; all these aspects are not usually met with synthetic conservation methods (nitrates, benzoates, sulfites, sorbates, formaldehyde) that despite being approved by government agencies continue to threaten health, by frequently inducing allergic reactions.

Some studies suggest that natural antimicrobials may be safer than synthetic ones; therefore, obtaining anti-infective agents from plants and algae can be an alternative strategy to develop new drugs that are safer, more effective and avoid bacterial resistance [99,100].

In general, the mechanisms of action of natural antimicrobials include disruption of the cytoplasmic membrane, inhibition of nucleic acid synthesis, decrease in proton motive force, and inhibition of energy metabolism (ATP depletion) [99,100]. Most antimicrobials derived from plants have been found in herbs and spices. These agents have different structural configurations that provide their antimicrobial action; the presence of hydroxyl groups (-OH) is believed to be the main cause of this property. Possibly due to the interaction of the -OH groups with the bacterial cell membrane that alters its structures and causes the leakage of its components. The antioxidant capacity is usually linked to the antimicrobial effect, which together (antioxidant/-OH groups) makes the compound more effective [99,100].

Currently, more than 1300 plants have shown antimicrobial activity, from which more than 30,000 compounds with this characteristic have been extracted. Plants and herbs (such as oregano, garlic, parsley, sage, coriander, rosemary, lemongrass, ginger, and chili), spices (cinnamon, cloves, curry, and pepper), and some essential oils (such as citral) have been shown to be effective against *Escherichia coli*, *Listeria monocytogenes*, *Campylobacter* spp., *Staphylococcus aureus*, *Salmonella* spp., *Pseudomonas aeruginosa*, *Vibrio cholerae* and *Bacillus cereus* [99,100,101,102,103]. Among the most relevant antimicrobial phytochemicals are thiosulfinates, glucosinolates, phenols, organic acids, flavonoids and saponins. However, those with the highest activity are phenols (terpenes, aliphatic alcohols, aldehydes, ketones, acids and isoflavonoids [99,100,101,102,103].

Although Ginestra et al., (2009) indicated that cladodes of *O. ficus indica* contain glucose, kaempherol and isorhamnetin, and apparently do not have antimicrobial activity, even after an enzymatic treatment; it was not a reason to rule out the development of further investigations [104]. In this sense, Sánchez et al., (2010) measured the synthesis of ATP, minimal bactericidal concentrations (MBCs), and changes in the integrity and potential of the *Vibrio cholerae* membrane after exposing it to methanolic, ethanolic and aqueous extracts of OFI var. Villanueva L. The three types of extracts were active against the bacteria (MBCs ranged between 0.5 and 3.0 mg/mL), were able to break cell membranes and cause an increase in their permeability [105].

These results opened the studies to the control of bacterial contamination in food and subsequently, the antimicrobial activity of non-polar extracts (petroleum ether and Chl) and polar extracts (MeOH and water) from the dried stems of OdHw and rhizome of *Zingiber officinale* were compared with *Bacillus subtilis*, *Staphylococcus aureus* and *Salmonella typhi*. The results also confirmed that this last bacterium is resistant to all extracts of both plants. Unlike *E. coli* and *B. subtilis* that were inhibited with the ether and chloroform extracts of OdHw, as well as with the MeOH and aqueous extracts of *Z. officinale*. These data suggest that the beneficial property of both plants is affected by the polarity of the extraction solvent [106].

In general, the studies developed to date (Table 5) suggest that different extracts (Hx, MeOH, EtOH, Chl, EtOAc, acetone (Ace), aqueous, dichloromethane (DCM) and mucilage) and/or oils from PPFs, CLDs (ripe and nopalitos), flowers, seeds and fruit peel especially obtained from OFI, OdHw, *O. xoconostle* (OX), *O. albicarpa* (OA), *O. stricta*, *O. microdasys* (OMs) and *O. macrorhiza* Engelm (OME) have shown antimicrobial action against Gram-positive bacteria (such as *Staphylococcus aureus*, *Staphylococcus haemolyticus*, *Listeria Monocytogenes*, *Bacillus cereus*, *Bacillus subtilis*, *Bacillus thuringiensis*, *Enterococcus faecalis*, *Streptococcus pneumoniae* and *Micrococcus* flavus) and Gram-negative bacteria (*Escherichia coli*, *Vibrio parahaemolyticus*, *Klebsiella pneumoniae*, *Pseudomonasaeruginosa*, *Pseudomonas fluorescens*, *Salmonella typhimurium*, *Acinetobacter lwoffii*, *Acinetobacter baumannii*, *Campylobacter coli*, *Campylobacter jejuni*, *Porphyromonas gingivalis*, *Prevotella intermedia*, *Enterobacter cloacae*, *Stenotrophomonas maltophilia*, and *Neisseria gonorrhoeae*) [107,108,109,110,111,112,113,114,115,116,117,118,119,120,121,122,123,124,125,126,127,128,129].

### 4.4. Action in the Treatment of Skin Wounds

The skin is the largest organ of the human body that acts as a protective barrier against harmful agents from the external environment. It controls thermal regulation and homeostasis of water and electrolytes. When this barrier is damaged, the body promotes the healing and/or scarring process to regenerate the injured area, involving molecular, cellular, and biochemical mechanisms that are divided into four phases (hemostasis, inflammatory, proliferative, and remodeling) (Figure 2). Any disruption in the balance of these processes causes problems and delays in wound healing; deteriorations related to aging, pathological situations (such as diabetes, obesity and/or arterial diseases) and multiple local and systemic factors (hypoxia, OXs, diminished immune response, poor nutrition, medications and infectious agents) [130].

Cicatrization is a physiological process that involves perfect interactions of numerous cells and molecules, so the imbalance of these interactions generates alterations during the process that are expressed as excessive yellow discharge, pain, swelling, redness and fever. Among the most relevant is the chronic inflammatory state, where pro- and anti-inflammatory mediators produce an exacerbated recruitment of neutrophils and macrophages with overexpression of inflammatory cytokines and excessive release of ROS, which together interfere with the proliferation/differentiation of keratinocytes and fibroblasts in the injured area and leads to cell apoptosis [130]. In addition, the increase in proinflammatory cytokines affects subsequent wound healing mechanisms, increasing matrix metalloproteinases (MMPs) and other proteases that alter cell proliferation/migration and reduce the accumulation of extracellular matrix components. Normally, there must be a balance between proliferation/activation and maturation/apoptosis of blood vessels; and if this is not done correctly, neovascularization and blood flow in the area are reduced, delaying the subsequent mechanisms of the proliferative and remodeling phase. Another mistake is the involvement of wound keratinocytes, which acquire a hyperproliferative state due to overexpression of the β-catenin/c-myc pathway, and express low levels of keratins 1, 2, and 10. This alters the migratory potential of these cells, which is related to the proteolytic degradation of growth factors and extracellular matrix proteins necessary for migration. Impaired remodeling is another major failure, as injured cells synthesize excessive amounts of MMPs and other proteases, degrading not only extracellular matrix components, but also cell surface receptors, growth factors, and the cytokines. In addition, inhibitors of metalloproteinases (TIMP) are reduced, contributing to the deregulation of proteases in these lesions, and consequently, the degradation of important molecules of the extracellular matrix such as collagen, elastin, fibronectin and chondroitin sulfate [130].

Over the years, adequate therapies have been sought to improve or promote the wound healing process. Currently, there are several treatments that can be classified into surgical procedures (autografts, allografts and xenografts), non-surgical therapies (topical formulations, dressings and skin substitutes) and pharmacological agents. However, depending on the size, type of wound, and factors that caused the damage, existing therapies are not completely effective [131]. Once again, phytomedicine, being popular among the general population in different regions of the world, opens new avenues of pharmacological intervention for the healing of skin wounds. Among the known phytotherapeutic agents are Aloe vera, mimosa (*Mimosa sensitive*), grape vine (*Vitis vinifera*), chamomile (*Matricaria chamomilla*), ginseng (*Panax ginseng*), jojoba (*Simmondsia chinensis*), rosemary (*Salvia rosmarinus*), lemon (*Citrus limon*), comfrey (*Symphytum officinale*), papaya (*Carica papaya*), oats (*Avena sativa*), garlic (*Allium sativum*), ginkgo (*Ginkgo biloba*), ocimum (*Ocimum basilicum*), tree oil, and olive oil [131].

In the case of *Opuntia* spp. and its extracts, there is various evidence of its use in traditional medicine for the treatment of burns, skin disorders and wound healing. The first study in this field of research compared the healing activity of a base cream containing lyophilized cladodes of OFI at 15% against a commercial ointment on wounds produced on the back of rats. After 5 days of treatment, the epithelialization process was evident and complete, suggesting that cladodes accelerate the proliferation and migration of keratinocytes in the cicatrization process [132]. The previous result was confirmed when two lyophilized polysaccharide extracts obtained from OFI were applied topically for 6 days and observed that they induce a beneficial effect (accelerate the re-epithelialization-remodeling phases and favor cell-matrix interactions) in the skin wounds of rats [133].

Using benzopyrene- or TNF-α-stimulated keratinocytes, Nakahara et al., (2015) demonstrated that CLD extracts protect the epidermal barrier and keratinocyte function by increasing the expression of filaggrin and loricrin, two proteins present in keratinocytes and corneocytes differentiated. In addition, they attribute the protective effect to an inhibition of ROS production caused by inflammatory agents. This property is probably related to the activation of nuclear erythroid factor (Nrf2) and NAD(P)H:quinone oxidoreductase 1 [134]. It is considered that the cicatrizant properties of OFI cladodes may involve high molecular weight polysaccharide components (such as linear galactan polymer and highly branched xyloarabinan) as well as low molecular weight components [lactic acid, D-mannitol, piscidic, eucomic, and 2-hydroxy-4-(4′-hydroxyphenyl)-butanoic acid]. These extracts could accelerate cell regeneration in a keratinocyte monolayer, which suggest that OFI components exhibit high anti-inflammatory and wound-healing properties [135].

Likewise, polysaccharides extracted from OFI stimulate the proliferation of fibroblasts and keratinocytes [136]. Among the protective agents present in *Opuntia* extracts, isorhamnetin glucoside components [such as isorhamnetin-glucosyl-rhamnoside diglucoside (IGR)], could inhibit COX-2, TNF-α and IL-6 production and induction of NO evoked by LPS [52].Not only OFI has shown beneficial effects, *O. humifusa* (OHF) extracts regulate the production of hyaluronic acid (HA) by increasing the expression of HA synthase in keratinocytes exposed to UV-B treatment. Treatment with these extracts could decrease the increased expression of hyaluronidase UV-B. The same protective effect on HA has been observed in SKH-1 hairless mice exposed to UV-B, which indicates that OHF extracts have a great capacity for skin care [137]. Table 6 presents all the studies of *Opuntia* spp. that justify its efficacy at the molecular and cellular level for the healing of skin wounds, as well as its use in dermatological preparations.

## 5. Toxic Evidence of the Genus *Opuntia*

The cacti family contains approximately 200 genera and 2000 species, which favors a wide genetic diversity that, together with environmental conditions (climate, humidity), type of soil, age of maturity of the cladodes and the harvest season, generates differences in the phytochemical composition of its vegetable parts (PPFs, CLDs, roots, flowers, seeds and stems) between wild and domesticated species, inducing changes in its nutritional values and undoubtedly in its functional and therapeutic properties. In this sense, although the public and some health professionals consider herbal medicines to be relatively safe because they are “natural”, there is very little data to support this assumption. Therefore, *Opuntia* spp. species are not exempt from possible adverse and toxic effects.

Saleem et al., (2005) evaluated for the first time its toxicological safety by determining the hypotensive activity of a methanolic extract of OdHw and its alpha-pyrone glycoside (opuntioside-I) in normotensive rats. At the end of their study, they observed no mortality with the extract and/or opuntioside-I orally administered, even at high doses of 1000 mg/kg/day. However, histopathological analysis revealed slight changes in the liver and spleen of the animals [145]. Subsequently, in 2012, the physicochemical characteristics (acidity, percentage of free fatty acids, saponification value, refractive index and density), lethal dose 50 (LD_50_) and toxicity of an OFI seed oil in mice were determined. Finding that the LD_50_ values ranged between 40.7 and 45.4 mL/kg body wt for oral administration and 2.52–2.92 mL/kg body wt for intraperitoneal administration [146].

These results and variations in doses called the attention to analyze other species of *Opuntia*, e.g., Osorio-Esquivel et al., (2012) who determined the acute toxicity of a MeOH extract of *O. xoconostle* (OX) seeds in mice fed with a hypercholesterolemic diet; finding that it was greater than 5000 mg/kg of body weight without the presence of apparent toxic manifestations [147]. Similar data on the absence of any sign of acute toxicity were observed in two other studies; the first, when orally administering up to 5 mL/kg of cactus pear seed oil (CPSO) to Wistar rats to determine its hypoglycemic effect [148] and/or by evaluating the in vitro and in vivo bioactivities of *O. macrorhiza* Engelm seed oil (OMESO) [60].

Considering that OFI is an important dietary source, a toxicological evaluation of aqueous extracts from different parts of the plant was performed and compared using three types of assays (MTT, Comet and the γH2AX In-Cell Western). The conclusion was that the fruit pulp extracts showed the best antigenotoxic effect against H_2_O_2_ and that no extract induced genotoxicity and/or cytotoxicity in the cell lines used [149].

To confirm the above findings, Han et al. (2019) investigated the genotoxicity of three doses of an OFIS extract (500, 1000 and 2000 mg/kg/day) orally administered for one week in rodents using the Ames test (*S. typhimurium* strains TA100, TA1535, TA98 and TA153 and *E. coli* strain WP2 urvA), chromosomal aberration assay in Chinese hamster lung cells and micronucleus test in bone marrow cells. In summary, it was observed that: (a) OFIS did not alter normal animal behavior or body weight gain, (b) mutagenicity was not present in both bacterial strains with or without S9 activation and (c) the number of micronucleated polychromatic erythrocytes (MPE) was not increased [150].

Recently, two groups of researchers addressed the safety of OdHw, considering that it is a cactaceae traditionally used in several countries to treat ailments such as inflammation, gastric ulcers, diabetes, hepatitis, asthma, and intestinal spasms. In the first study, the acute toxicity of the oil obtained from its seeds was evaluated in albino mice and Wistar rats. After a single administration of the established doses (1.0, 2.0, 3.0, 5.0 and 7.0 mL/kg), adverse signs and/or mortality were observed for four weeks. The conclusion was that the oil produced no variations in the body weight of the animals and no mortality or signs of toxicity during the entire monitoring period. In addition, cell viability was not affected when human hepatoma HepG2 culture was analyzed [151,152].

Finally, when evaluating a MeOH extract of cladodes by MTT assay in human embryonic kidney cell line, genomic DNA fragmentation using agarose gel electrophoresis and bone marrow micronuclei frequency, it was proved that a 7-day treatment of 5 g/kg of the extract orally had no effect on DNA integrity, neither did it induce cytotoxicity or stimulate MPE formation [153]. Unfortunately, although *Opuntia* spp. could be considered a reliable and safe plant, some authors have identified and reported the presence of certain side effects during oral consumption of OFI, such as mild diarrhea, increased stool volume and frequency, nausea, headache and lower colonic obstruction [6,20,103,149].

Despite these secondary effects, plants of the *Opuntia* genus are traditional foods frequently consumed and their cladodes and fruits are still considered with high agrotechnological potential. Besides, studies suggesting an LD₅₀, above 5000 mg/kg are safe levels. To date, there is no established dose and/or concentration for its consumption and there are different intervals that depend on the route of administration, the species (humans/animals) in which they are used; the approach to use it whether it is food (fresh, juices or extracts) or for experimental evaluation.

Some authors recommend an intake between 10 to 17 g/person/day of *Opuntia* and/or prickly pear fruits (PPFs) to have a healthy life. Others suggest between 100 and 500 g/day of roasted CLDs to significantly reduce the complications of diabetes mellitus. There are products, such as PPFs, in commercial presentations of capsules, tablets, powders, and juices whose oral dosage regimens are established at 250 mg/3 times a day/every 8 h [18,154,155].

In general, summarizing the information from both documents (part 1 and 2) it can be seen that the doses range from 50 mg/kg to 7 g/kg [23].

## 6. Conclusions and Perspectives

Although modern medicine is available in most countries for the control and treatment of many diseases, phytomedicine and/or TCAM continue being popularly used in different populations for historical, cultural reasons, easy access, low cost, diversity, and especially, a relative lower quantity of adverse effects. The set of studies presented in both reviews (Part 1 and 2) demonstrate the beneficial properties of the different vegetative parts of *Opuntia* spp. (wild and domesticated). For this reason, scientific research on this genus of plants (known as succulents, due to their ability to generate biomass by storing water in one or more of their organs) has deepened and may continue to increase, in order to better understand their nutritional and therapeutic properties.

In general, most of the evidence confirms that CLDs, PPFs, oils and/or extracts (MeOH, Hx, EtOAc, Chl and aqueous) coming mainly from OFI, OS, OdHw, OHF, OX and *O. macrorhiza* Engelm have presented relevant therapeutic and/or pharmacological potentials; whose mechanisms of action are mainly related to the inhibition of the absorption of substances, modification of the intestinal flora, elimination of reactive oxygen species and/or protection of nucleophilic DNA sites, anti-inflammatory activity, induction of detoxification pathways, and activation of apoptosis. However, it is convenient to extend the investigations to other species in order to analyze and confirm their pharmacological capacities.

Likewise, the results of the investigations confirm and coincide that these beneficial properties are possible attributed to a synergistic and/or combined effect among the different bioactive compounds (vitamins, flavonoids (isorhamnetin, kaempferol, quercetin), phenolic compounds, pigments (carotenoids, betalains and betacyanins), α-pyrones (opunthiol and opuntioside glucoside), pectin and mucilage). Nonetheless, it is convenient to increase the individual studies of each phytochemical, to determine its protective action; given that as substrates they can activate different biochemical reactions to provide important health benefits and be recognized as significant nutraceutical agents. All of the above, added to the fact that several *Opuntia* plants have been consumed by humans for more than 8000 years, which are easily adapted and/or propagated in different types of soil. In addition to their relatively low presence of adverse and toxic effects, they favor their domestication process, the increase in economic interest and new advances in the field of biotechnology.

It is convenient to remember that the process to discover drugs and/or medications is complex and costly. In that process different types of studies converge (such as those presented in this document; in vitro, in vivo and clinical). In recent years, computational methods (also called in-silico) have been integrated into this multidisciplinary effort, contributing to efficient data analysis, filtering and/or selecting individual bioactive molecules for their subsequent experimental evaluation, and also to generate hypotheses that favor the understanding of its mechanism of action and the design of new chemical structures.Again, *Opuntia* species are not exempt from participating in this area of research. Among the most significant studies, those carried out by Elkady et al., (2020), who isolated and characterized constituents of the prickly pear peel to determine their antibacterial activity stand out. This latter assay revealed that quercetin 5,4′-dimethyl ether found in EtOAc fraction exerted an inhibitory effect against pneumonia pathogens. Virtual docking of the isolated compounds showed promise in silico anti-quorum sensing efficacy, suggesting that unused waste from fruits contains bioactive components with possible beneficial potential [126]. On the other hand, an In Silico Investigation on the Interaction of Chiral Phytochemicals from *O. ficus-indica* with SARS-CoV-2 Mpro (main viral protease) was developed. Using two web-based molecular docking programs (1-Click Mcule and COVID-19 Docking Server) several flavonols and flavonol glycosides were identified; highlighting the chiral compound astragalin with high binding affinity for Mpro and a low toxicity profile. Emerging the possibility of a protease inhibitor agent as an anti-COVID-19 strategy [156]. In the most recent study, the possible targets in the PI3K/Akt/mTOR pathway acted upon by an *O. xoconostle* extract were modeled and simulated in silico using the Big Data-Cellulat platform, as well as the concentration range of LD₅₀ to be used in breast cancer cells. The in silico results showed that the activation of I3K and Akt is related to angiogenesis and inhibition of apoptosis, and that the extract has an antiproliferative effect on cancer cells, causing the cells to interrupt in the G2/M phase of the cell cycle [157]. Taken together, these three studies demonstrate that the use of in silico tools is a valuable method for conducting virtual experiments and discovering new therapeutic agents.

In conclusion, there is still a long way to go on scientific research to understand in more detail the significant beneficial properties of all species of *Opuntia* spp.

## Figures and Tables

**Figure 1 plants-11-02333-f001:**
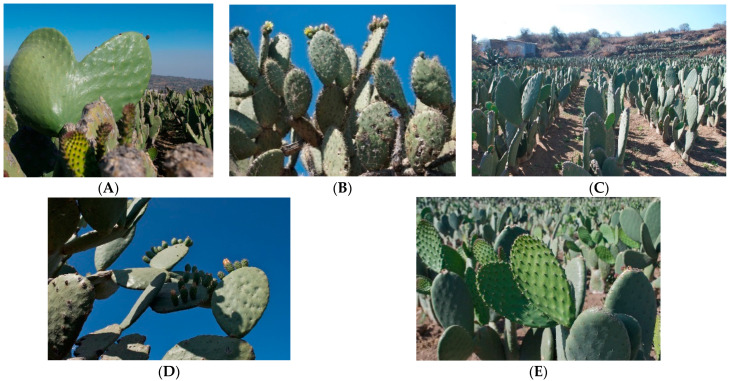
Mexican cultivars of *O. Streptacantha* (**A**), *O. megacantha* (**B**) and *O. ficus-indica* (**C**), *O. hyptiacantha* (**D**), and *O. albicarpa* (**E**).

**Figure 2 plants-11-02333-f002:**
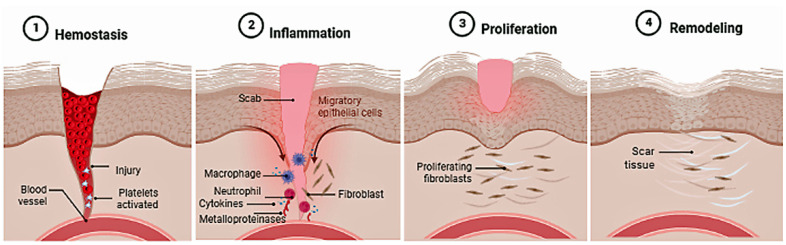
Stages of skin healing.

**Table 1 plants-11-02333-t001:** Main products and by-products obtained from *Opuntia* (Nopal).

Products from Cladodes (CLDs)	Products from Prickly Pear Fruits (PPFs)	By-Products from CLDs and PPFs
Lacto-fermented pickles	Juices, nectars, pulps, purees	Fruit and seed oil
Candies, sweets	Jams, jellies	Peel pigments and/or fruit debris
jams	fruit leather and teas	Cladode extracts
flours	syrup, sweetener	Dietary fiber and cladode mucilage
Fresh and cooked vegetables	Bioethanol, wine	Pigments for cosmetics
Ethanol	Canned and frozen fruit	
Edible coating	Juice concentrates and spray dried juice powder	

Table modified from Feugang et al. (2006) [18].

**Table 2 plants-11-02333-t002:** Nutritional composition in different anatomical parts of *Opuntia ficus-indica* (L.) Mill.

Chemical Species	Main Component
Cladodes	
Minerals	K and Ca (mainly calcium oxalate crystals).
Vitamins	E, A, C, B1, B2, B3
Amino acids	Glutamine, arginine, leucine, isoleucine, lysine, valine and phenylalanine
Fatty acids	Palmitic acid, oleic acid, linoleic acid and linolenic acid
Carotenoids	Lutein, β-carotene and β-cryptoxanthin
Flavonoids	Quercetin, kaempferol, isoquercetin, isorhamnetin-3-O-glucoside, nicotiflorin, rutin
Phenolic compounds	Coumaric Gallic acid, 3,4-dihydroxybenzoic 4-hydroxybenzoic, and ferulic acid
Prickly pear fruits	
Minerals	K, Ca, and Mg
Vitamins	E, A, and C
Amino acids	Lysine, methionine, glutamine, and taurine
Organic acids	Maleic, malonic, succinic, tartaric, and oxalic
Pigments	Betaxanthins, betacyanins, and betalains
Fatty acids	Palmitic acid and linoleic acid
Flavonoids	Kaempferol, quercetin, and isorhamnetin
Seeds	
Minerals	K and P. Lower proportions of Mg, Na and Ca
Sterols	β-sitosterol and campesterol
Fatty acids	Palmitic acid, oleic acid, and linoleic acid
Phenolic compounds	Ferulic acid, sinapoyl-diglucoside, synapoyl-glucose, and feruloyl-sucrose
Pulp and peel	
Minerals	K, Ca and Mg
Sterols	β-sitosterol and campesterol
Fatty acids	Palmitic acid, oleic acid, linoleic acid, stearic acids and linolenic acid
Carotenoids	Lutein, β-carotene, violaxanthin, lycopene, and zeaxanthin
Flavonoids	Quercetin, isorhamnetin, kaempferol, luteolin, and isorhamnetin glycosides
Phenolic compounds	Ferulic acid, sinapoyl-diglucoside, and feruloyl-sucrose isomer
Flowers	
Flavonoids	Kaempferol, quercetin, and isorhamnetin glycosides
Organic acids	Mainly gallic acid

Table modified from Madrigal-Santillán et al. (2022) [23].

**Table 3 plants-11-02333-t003:** Studies testing for anti-inflammatory and antiulcerative effects of *Opuntia* spp.

Type of Study	Objective and Characteristics	Results and Conclusion	Ref
**Anti-Inflammatory Evidence**			
In vitro	Considering that the inflammatory response depends on the redox state of an organism and the evidence of antioxidant properties of some OFI pigments, the protective effect of betalains on vascular endothelial cells as a direct target of OXs in inflammation was analyzed.	The result indicated that betalains protect the endothelium from oxidative alteration through the inhibition of some cytokines (such as ICAM-1).	[41]
In vivo	Using the rat paw edema model induced by CRRG, the anti-inflammatory effect of alcoholic extracts of flowers, fruits and stems from OdHw was evaluated. To analyze the analgesic potential of the same extracts, electric current was used as a noxious stimulus.	An important observation was that the flower extract (dose of 200 mg/kg) had the highest anti-inflammatory and analgesic capacity. Furthermore, by performing a bioassay-guided division by using VLC, Sephadex, and paper chromatography, three flavonoid glycosides (kaempferol 3-O-alpha-arabinoside, isorhamnetin-3-O-glucoside, and isorhamnetin-3-O-rutinoside) were obtained and related to its protective capacity.	[42]
In vitro	Hypochlorous acid (HOCl) is produced from H_2_O_2_ and chloride by the enzyme heme- myeloperoxidase (MPO). It is a powerful oxidant produced by Neut that contributes to the damage caused by these inflammatory cells. The objective of the study was to analyze the interaction of betanin and indicaxanthin (Ind) with HOCl and compounds I and II of MPO.	The conclusion was that both betalains were good substrates for MPO and function as one-electron reducing agents of their redox intermediates (compounds I and II). Moreover, the two pigments effectively removed HOCl at 25 °C with a pH between 5 and 7.	[43]
In vitro	The aim of the study was to determine the antioxidant potential and the anti-inflammatory effect in macrophages (RAW264.7 cells) producers of nitric oxide (NO) of different extracts (MeOH, Hx, Chl, EtOAc, butanol and water) prepared from the leaves of OHF	Using the 2,2-diphenyl-1-picrylhydrazyl (DPPH) scavenging assay, the presence of antioxidant activity in all the extracts was confirmed and the EtOAc fraction was the most significant. Regarding the anti-inflammatory effect, only Chl and EtOAc fractions suppressed the production of NO in RAW264.7 cells activated by lipopolysaccharide (LPS). A significant inhibition of the expression of inducible nitric oxide synthetase (iNOS) and interleukin-6 (IL-6) was also evidenced; hence, OHF can modulate the expression of inflammatory cytokines.	[44]
In vitro	Given that the conventional medications reduce symptoms of osteoarthritis (OA) but can cause significant side effects, the use of natural substances that reduce and/or delay the progression of that disease has begun to be explored. Therefore, the anti-inflammatory effect of some lyophilized extracts obtained from OFI cladodes on the production of NO, glycosaminoglycans (GAGs), prostaglandin G2 (PGE2), and ROS in cultured human chondrocytes stimulated by interleukin-1 beta (IL-1β) was analyzed.	The DPPH assay showed that the freeze-dried substances have a significant antioxidant effect. Besides, the results indicated that all extracts counteracted the deleterious effects of IL-1β by decreasing the production of key molecules released during the chronic inflammation. These data suggest that the extracts exert a chondroprotective capacity greater than that caused by hyaluronic acid commonly used as visco-supplementation in the treatment of OA.	[45]
In vitro	In order to increase the scientific information of OFI on its beneficial properties in the inflammatory response, the immunomodulatory activity of an aqueous extract supplemented to a mouse Macrop culture was studied.	The cells were cultured in RPMI-1640 to analyze the presence of NO, iNOS and NF-κB induced by LPS. The results of the immunosorbent assay and Western method showed that all inflammatory parameters were suppressed when adding the extract to the culture. This opens the possibility of considering the aqueous extract as a nutraceutical ingredient applicable to functional foods.	[46]
In vitro	Evidence from in vitro and in vivo studies with OHF suggests anti-cancer and anti-inflammatory activity on different cancer cells. Therefore, to confirm these biological properties, different OHF extracts were obtained and evaluated by the DPPH assay as well as analyzed their bioactive fractions to determine the cytotoxicity on human colon cancer (SW480) and breast cancer (MCF7) cells.	The EtOAc extract showed the highest cytotoxicity and regulated the expression of the proapoptotic protein Bax (bcl-2-associated X protein) in both cell lines. Likewise, the incubation of cells with this extract reduced the induction of inflammatory molecules (COX-2 and iNOS) mainly in SW480 cells. These results indicate that these cells are more susceptible to the bioactive compounds of the extract, which may be potentially preventive to cancer by modulating apoptotic markers and inhibiting inflammatory pathways.	[47]
In vitro	This study demonstrated the anti-inflammatory activity of Ind in an IBD model (i.e., intestinal Caco-2 cells stimulated by IL-1β; which induces ROS and iNOS to activate NF-κB and trigger the release of proinflammatory mediators).	Coincubation of cells with Ind (concentrations from 5 to 25 μM) prevented the release of proinflammatory cytokines (IL-6, IL-8, PGE2 and NO) in a dose-dependent manner. The expression of COX-2 and iNOS was also reduced. In conclusion, the findings indicate that betalain could modulate inflammatory processes in the intestine.	[48]
In vivo	Using a rat model with acute inflammation, the protective activity of Ind from OFI was evaluated. Pleurisy was induced by injecting 0.2 mL of CRRG into the pleural cavity. Subsequently, the animals were sacrificed (4, 24 and 48 h) to collect exudates and analyze different inflammatory parameters.	Prior oral administration of Ind (0.5, 1, and 2 μmol/kg) decreased the volume of exudate and the number of leukocytes in the pleural cavity (95%). Likewise, the highest dose of Ind inhibited the expression of PGE2, NO, IL-1β, COX-2, iNOS and TNF-α. These results suggest that the pigment has the potential to improve health and prevent inflammatory disorders.	[49]
In vitro	In this research, natural flavonoid-rich concentrate (FRC) extracted from OFI juice was tested on intestinal inflammation induced in Caco-2 cells. Through an adsorption separation process, the FRC was obtained and its main components (isorhamnetin 3-O-rhamnose-rutinoside, isorhamnetin 3-O-rutinoside, and ferulic and piscidic acid) were identified. Subsequently, its effect (coincubation or preincubation) on the EOx induced by H_2_O_2_ in these human cells was evaluated.	Results showed that coincubation significantly attenuated ROS production; suggesting that bioactive compounds cannot freely cross the cell membrane. A similar phenomenon occurred in the inflammatory response, achieving a decrease in IL-8 secretion. FRC also reduced the expression of NO and TNF-α; however, there were no differences between pre and coincubation.	[50]
In vivo	Using the tail twisting and dipping test (Tail Flick) and the CRRG-induced paw edema test performed in albino Wistar rats, the antinociceptive and anti-inflammatory action of prickly pear juice extracted from *O. elatio* Mill was proved.	The results confirmed that the ED_50_ of the juice is between 0.919 and 9.282 mL/kg and that both pharmacological effects occur in a dose-dependent manner. Possibly, the betacyanins in the juice are responsible for exerting these potentials.	[51]
In vitro In vivo	Considering the biological properties of OFI attributed to different phytochemicals present in its composition, the effect of an extract and its main isorhamnetin glycosides on some inflammatory markers in vitro and in vivo was evaluated. Initially, the extract was obtained by alkaline hydrolysis, while the bioactive compounds were purified by preparative chromatography.	Using the croton oil-induced ear edema model, the expression of inflammatory markers was determined. The conclusion of the study is that the diglycoside isorhamnetin-glucosyl-rhamnoside (IGR) was the most significant bioactive compound. Both IGR and the extract suppressed the expression of NO, COX-2, TNF-α; IL-6; especially, NO production was decreased in LPS-stimulated RAW 264.7 cells.	[52]
In vitro	Filannino et al., (2016) studied the ability of lactic acid bacteria (*Lactobacillus plantarum* CIL6, POM1 and 1MR20, *L. brevis* POM2 and POM4, *L. rossiae* 2LC8 and *Pediococcus pentosaceus* CILSWE) to increase the antioxidant and anti-inflammatory potential of the pulp of OFI cladodes in intestinal Caco-2/TC7 cells. The pulp was fermented with the different strains of bacteria isolated from fruits and vegetables and the flavonoid profile was defined at the end of the study.	In conclusion, fermentation with *L. plantarum* and *L. brevis* favored the highest concentration of γ-amino butyric acid and had preservative effects on vitamin C and carotenoid levels. Using Caco-2/TC7 cells and after inducing oxidative EOx by IL-1β, an increase in antioxidant activity and an immunomodulatory effect was confirmed in the presence of the same bacteria. Kaempferol and isorhamnetin were identified as the main compounds responsible for the increase in radical scavenging activity.	[53]
In vivo	*O. dillenii* (Nagphana) is a plant native to Central America traditionally used for its analgesic and anti-inflammatory effects. Unfortunately, there is little scientific evidence to support these properties. Therefore, in order to evaluate these properties, 12-O-tetradecanoyl-phorbol-13-acetate (TPA)-induced ear edema accompanied by histological studies of mice ear sections and phospholipase A2 (PLA2)-induced mice paw edema were used, parting from a MeOH extract. In parallel, levels of leukotriene B4 (LTB4) and ROS were also determined via HPLC and the levels of PGE2, TNF-α, IL-1β and -6 were measured by ELISA assay. Finally, its main α-pyrones [opuntiol (aglycone) and opuntioside glucoside (O-glucoside)] were isolated by vacuum liquid chromatography.	Both the extract and the α-pyrones reduced TPA-induced ear punch weight in a dose-dependent manner. IC_50_ values demonstrated a suppression of inflammatory features histologically observed. In addition, paw edema and peritonitis were also attenuated. In comparison to indomethacin and diclofenac sodium, opuntioside reduced PGE2 levels in the inflamed ear, which was 1.3 times better than opunthiol. However, opunthiol was more potent in reducing LTB4 levels in rat neutrophils and effectively suppressed ROS and cytokine (TNF-α, IL-1β and -6) levels. In general, the data justify the traditional use of *O. dillenii* and suggest that its CLDs possess anti-inflammatory properties through the inhibition of arachidonic acid metabolites and cytokines. Likewise, opunthiol can be considered a COX-2 inhibitor. However, opuntioside showed its selectivity towards PGE2 without affecting LTB4 levels.	[54] [55]
In vivo	The results obtained by Cho et al. [44] and Kim et al. [47] motivated to design this study to reveal the anti-nociceptive and anti-inflammatory effect of a MeOH extract of OHF stem. The first potential was evaluated by hot plate, acetic acid (AcOH)-induced writhing, and tail-flick assays in mice and rats. In contrast, the anti-inflammatory capacity was measured in tests of vascular permeability and paw edema induced by CRRG and serotonin. To confirm this effect, it was also measured using LPS-induced RAW 264.7 cells.	The extract inhibited writhing and delay reaction time of rodents to hot plate-induced thermal stimulation and tail-flick tests. Similarly, paw edema induced by CRRG and serotonin was attenuated. Evans blue concentration was significantly decreased in the vascular permeability test, confirming a strong anti-inflammatory effect. Finally, the n-butanol fraction reduced the expression of iNOS and NO in RAW 264.7 cells.	[56]
In vivo In vitro	The purpose of this study was to evaluate the antigenotoxic (using the Allium cepa test), analgesic and anti-inflammatory properties of *O. microdasys* in the post-flowering stage F3 (OMF3) by means of tests similar to previous studies.	The dose of the aqueous extract (100 mg/kg) decreased the writhing induced by AcOH and the edema produced by CRRG. Likewise, OMF3 had an antimutagenic potential against DNA damage mediated by H_2_O_2_.	[57]
In vivo	The fruit vinegars (FVs) available in Algeria are used in folk medicine for their hypolipidemic properties. The preventive effects of three types of FVs (PPFs from OFI, pomegranate and apple) against obesity-induced cardiomyopathy and their possible mechanisms of action were studied. Wistar rats on a high-fat diet (HFD) were treated with AcOH and different doses (3.5, 7.0 and 14.0 mL/kg) of the three types of FVs for 18 weeks. Subsequently, plasmatic biomarkers of inflammatory and cardiac enzymes were evaluated.	At the end of the treatment, the FVs decreased the increase in body weight induced by HFD, as well as the increase in plasma levels of fibrinogen and leptin. Furthermore, these treatments preserved the myocardial architecture and attenuated the cardiac fibrosis. These findings suggest that FVs (especially PPFs) can prevent obesity and its HFD-induced cardiac complications; This prevention is probably related to its anti-inflammatory and hypolipidemic properties.	[58]
In vivo	In this study, we did a research on the effects of polyphenol-rich infusions of OFI cladodes on obesity-associated inflammation and ulcerative colitis induced by dextran sulfate sodium (DSS) in Swiss HFD mice. For 4 weeks, the animals received an infusion of 1.0% CLDs and were subjected to the administration of DSS for the following 7 days. At the end, the rodents were sacrificed to determine the levels of proinflammatory cytokines.	The infusion decreased the severity of inflammation associated with obesity and acute colitis exerted by DSS. The expression of TNF-α, and IL-6 in the colon and spleen was significantly reduced. Therefore, the anti-inflammatory potential of the infusion could be attributed to the polyphenols present in its chemical composition.	[59]
In vivo In vitro	The food industry maintains a continuous search for ingredients that provide beneficial properties to its products, whether considering their nutritional value, bioactivity, flavoring and/or technological aspects. The crude oil from the seed of *O. macrorhiza* Engelm (OMESO) is an ideal candidate for this type of ingredient, so it was chemically characterized and its in vitro and in vivo bioactivities were determined.	OMESO presented a low acidity index and oxidation stability; properties that favored its antioxidant and α-glucosidase inhibitory activity. In addition, it presented anti-inflammatory, analgesic and antibacterial potential (mainly against Gram-positive bacteria) and did not show any signs of acute toxicity in animals. This highlights its possible use in different food applications.	[60]
In vitro	The purpose of the study was to analyze the antioxidant and anti-inflammatory properties of OFI cladode extracts in BV-2 microglia cells. The inflammation associated with the activation of microglia in neuronal injury was achieved by exposing it to LPS and revealing its action on fatty acid β-oxidation and antioxidant enzymes in peroxisomes.	The different extracts showed an antioxidant effect through microglial catalase and an anti-inflammatory effect by reducing the production of NO LPS-dependent. These results suggest that the extracts have a neuroprotective activity through the induction of peroxisomal antioxidant activity.	[61]
Clinical study	Various scientific evidences have shown that products containing extracts of fruits and/or OFI cladodes have been favorably used to control obesity, lipid profile and glycemia. Therefore, the beneficial potential of a paste added with 3% of CLDs extract to human health was analyzed. A study was developed with 42 healthy volunteers who were administered 500 g/week of this paste for 30 days.	The paste demonstrated hypoglycemic, antioxidant and anti-inflammatory properties with a supposed effect on the aging process. Although the results were preliminary, there is a strong possibility that the paste would be considered an effective food for the prevention of some metabolic diseases.	[62]
Clinical study	As has been shown, dietary ingredients and food components are important factors in stimulating the immune system and preventing chronic inflammation responsible for some age-related diseases. In this randomized 2-period (2-week/period), controlled-feeding study involving 28 healthy patients, a diet supplemented with PPFs pulp (200 g/twice daily) on inflammatory plasmatic markers was explored.	At the end of the period, there was a reduction in proinflammatory markers (TNF-α, IL-8, IL-6, IL-1β, interferon-γ, and C-reactive protein (CRP)). Likewise, an increase in dermal carotenoids (“skin carotenoid score”, a biomarker of the antioxidant status of the human body) was established. The observed modulation of both inflammatory markers and antioxidant balance suggests that prickly pear may be a beneficial food for human health.	[63]
In vivo	The purpose of the study was to characterize the phytochemical composition of a hydroalcoholic extract from PPFs seeds by High-Performance Liquid with Diode-Array Detection (HPLC-DAD) analysis and to evaluate its anti-inflammatory and/or analgesic activity in rodents using again the edema assay of legs induced by CRRG, the resistance exerted by AcOH and the tail dip test.	The extract dose (500 mg/kg) showed a significant increase in the mean latency of the TAIL FLICK test and a decrease in the mean number of twisting movements in the KOSTER test. As well as an important anti-inflammatory activity in the pattern of paw edema. Which suggests that OFI seeds may be a possible natural source of new active ingredients with therapeutic action.	[64]
In vivo	Because different plants have been used as a source of effective and safe alternative of therapeutic agents for various ailments, the topical anti-inflammatory and antioxidant potential of pumpkin seed oil (*Cucurbita pepo*), flaxseed (*Linum usitatissimum*) and OFI were compared using the CRRG plantar edema test, hematological and biochemical analysis, EOx test and histological study.	All oils proved to be effective against acute inflammation. Animals treated, especially with OFI, revealed a significant decrease in hematological parameters (white blood cells and platelets) and concentrations of CRP and fibrinogen. Another observation was an increase in the activity of glutathione peroxidase (GPx), catalase (CAT) and superoxide dismutase (SOD) in the skin by reducing lipid peroxidation. The suggestion is that the anti-inflammatory effect of oils is related to the antioxidant properties of their bioactive compounds (polyunsaturated fatty acids, vitamin E and phytosterols).	[65]
In vitro	The study investigated betalain-rich extracts as a promising strategy for intestinal inflammation management. After obtaining the prickly pear betalain-rich extracts by means of a QuEChERS method and characterizing it by LC-DAD-ESI-MS/MS analysis, mainly betanin and indicaxanthin were found.	The extracts showed potent antioxidant and anti-inflammatory activities. Significant inhibition of ROS and inflammatory markers (IL-6, IL-8 and NO), even greater than dexamethasone, was observed in an in vitro model of IL-1β-induced intestinal inflammation.	[39]
In vivo	Due to the fact that exposure to Particulate matter (PM) can cause respiratory disorders. It was evaluated the protective effect of various extracts (water, ethanolic 30 and 50%) from OFI on airway inflammation associated with exposure to PM10D (diameter aerodynamic less than 10 μm). BALB/c mice were exposed to PM10D via intranasal tracheal injection three times over a period of 12 days and the extracts were administered orally for the same time.	All extracts suppressed neutrophil infiltration and the number of immune cells (CD3^+^/CD4^+^, CD3^+^/CD8^+^, and Gr-1^+^/CD11b) in bronchoalveolar lavage fluid and lungs. They also decreased the expression of cytokines, TNF-α, COX-2 and different interleukins (IL-17, IL-1α, IL-1β, IL-5, IL-6). These results suggest that OFI extracts may be used to prevent and treat respiratory diseases.	[66]
**Antiulcer evidence**			
In vivo	The effects of dry stem powder of OFI var. Saboten (OFIS) were studied in models of gastric ulcers and injuries induced by ethanol (EtOH) and acetylsalicylic acid (ASA), respectively.	OFIS showed a significant inhibition at doses of 200 and 600 mg/kg for gastric lesions produced by both chemical compounds (EtOH and ASA). In the same sense, it was determined that OFIS do not affect gastric juice secretion, acid production and pH. Which indicates that they only have an inhibitory action on gastric injury without anti-ulcer activity.	[67]
Clinical study	Some evidence suggest that the severity of alcoholic hangovers is related to inflammation induced by impurities in the beverage and byproducts of EtOH metabolism. Therefore, to try to reduce it, natural agents have been used. In this double-blind, crossover trial, 64 volunteers consumed an OFI extract (1600 IU) 5 h before ingesting 1.75 g/EtOH/Kg for 4 h. At the end of the period, symptoms of alcoholic hangover were evaluated and blood and urine samples were obtained.	In general, the symptom index was reduced; especially, the presence of nausea, dry mouth and anorexia. It was also observed that CRP levels were associated with the hangover severity; which decreased with the previous intake of OFI. These data confirm that hangover symptoms are mainly due to the activation of inflammation and that OFI has a moderate effect in reducing hangover symptoms.	[68]
In vivo	Considering the previous studies of Wiese et al. [68] as well as that OFI mucilage can slow down the rate of digestion and/or intestinal absorption, its effect (5 mg/kg per day) on the healing of gastritis induced by EtOH was evaluated in rats. Consequently, the lipid composition, expression of 5′-nucleotidase (membrane-associated ectoenzyme), cytosolic activity of alcohol dehydrogenase (ADH) in the gastric mucosal plasma membrane were determined. In addition, a histological analysis was included.	Results showed that EtOH induced loss of surface epithelium and infiltration of polymorphonuclear leukocytes. The activity of ADH and phosphatidylcholine (PC) diminished and the content of cholesterol in plasma membranes increased. In contrast, the administration of mucilage rapidly corrected these enzymatic changes and restored histological alterations and also reduced the damage to the plasmatic membranes of the gastric mucosa; showing an anti-inflammatory effect. Therefore, the beneficial action of the mucilage seems to be correlated with the stabilization of the plasmatic membranes of the damaged gastric mucosa. Molecular interactions between its monosaccharides and membrane phospholipids favor the healing process.	[69]
In vivo In vitro	Some phytochemical analyzes of the MeOH extract from the root of *O. ficus-indica* f. inermis (MEROfi) agree that it is rich in flavonoids and phenols, which is why its antioxidant activity was quantified in vitro (DPPH assay) and its gastroprotective capacity against EtOH-induced ulcer was evaluated in vivo.	The antiradical activity showed an EC_50_ of 119 μg/mL. Whereas the pretreatment of three doses of MEROfi (200, 400 and 800 mg/kg) reduced the ulcerative lesion at a rate of 82, 83 and 93% respectively. It also prevented the depletion of antioxidant enzymes, SOD, CAT and GPx.	[70]
In vivo In vitro	Since MEROfi demonstrated a positive effect on the ulcerative lesion [70], now we studied the same antiulcer action, antioxidant activity and reducing power of a 50% MeOH extract of the flowers obtained from the same species of *Opuntia* (MEFOfi).	Animals pretreated with MEFOfi (250, 500 and 1000 mg/kg) evidenced a dose-dependent protection against gastric damage caused by EtOH, avoiding deep epithelial necrotic lesions. The reduction of the ulcerative lesion was accompanied by an inhibition of lipid peroxidation, protein oxidation and restitution of the enzymatic activity of SOD and CAT. Possibly, the antiulcerogenic activity is attributed to a synergistic effect of antioxidant and antihistamine type.	[71]
In vivo	In this third approach, the research group of Alimi et al., (2010, 2011) explored the efficacy of administering two doses (2 and 4 mL/100g p.v.) of a fruit juice of *O. ficus indica* f. inermis (FJOfi) for 90 days on the reversal of oxidative damage induced by chronic EtOH intake in Wistar rat erythrocytes and with the use of HPLC, they determined the content of phenols and flavonoids.	HPLC analysis revealed high concentrations of phenolic acids and flavonoids. On the other hand, EtOH markedly decreased the activity of SOD and CAT. These changes in the antioxidant capacity of erythrocytes were accompanied by a greater oxidative modification of lipids and proteins (increase in carbonyl groups). In contrast, both doses of FJOfi significantly reversed decreases in enzymatic and non-enzymatic antioxidant parameters in erythrocytes. The protective effect of FJOfi highlights the inhibition of free radical chain reactions induced by EtOH.	[72]
In vivo	In order to develop this study, pectin was isolated from peel (WNPE) and pulp (WNPU) of *O. microdasys* var. rufida’s (OMR) and its main polysaccharides were characterized by gas chromatography coupled to mass spectrometer (GC-MS), nuclear magnetic resonance (1H NMR) and Fourier transform infrared spectroscopy (FTIR). Subsequently, the biopolymers were administered intraperitoneally (50-100 mg/kg) to mice to determine their gastroprotective, analgesic and anti-inflammatory effects.	The results showed that WNPE and WNPU are mainly composed of uronic acids and neutral sugars (such as arabinose, galactose, rhamnose, and mannose). A significant gastroprotective effect was observed with both biopolymers, reducing the presence of gastric ulcer at a dose of 100 mg/kg (between 67 and 82%). Regarding the analgesic and anti-inflammatory action, tests with chemical stimuli (writhing test) and thermal stimuli (hot plate test) determined a dose-dependent effect (especially between 50 and 100 mg/kg)	[73]
In vivo	The objective was to evaluate the protective effect of *O. dillenii* Haw fruit juice. (FJOdHw) on AcOH-induced ulcerative colitis in rats. FJOdHw was administered orally for 7 consecutive days before inducing colitis on the eighth day. Subsequently, biochemical tests and histopathological examinations of the colon were performed to assess the damage.	Pretreatment with FJOdHw (2.5 and 5 mL/kg) attenuated macroscopic damage and showed significantly reduced levels of myeloperoxidase, malondialdehyde, and serum lactate dehydrogenase. The results suggest that the antiulcer effect may be due phenols, flavonoids and betalains present in FJOdHw.	[74]
In vivo	As shown in the evidence included in this table, the different OFI extracts have been used in traditional folk medicine for various purposes, including action on inflammatory processes. This last experiment explored the prophylactic effect of an OFI fruit peel petroleum ether extract (FPPEE) against irradiation-induced colitis in rats.	A previous analysis with gas chromatography and mass spectrometry (GC/MS) identified 33 compounds in the unsaponifiable fraction and 15 fatty acid methyl esters in the saponifiable part. Of these, 13 terpenes and sterols were isolated; and 10 of their compounds had not been isolated before from any part of this species. On the other hand, FPPEE pretreatment decreased elevated levels of MPO, NO, COX-2, TNF-α, NF-κB, IL-10, and SOD. The conclusion was that FPPEE can limit the colonic complications generated by irradiation, possibly due to its antioxidant and anti-inflammatory properties.	[75]

**Table 4 plants-11-02333-t004:** Scientific evidence of the neuroprotective effect of *Opuntia* spp.

Type of Study	Objective and Characteristics	Results and Conclusion	Ref
In vitro	The author examined the inhibitory action of two concentrations of OFI methanolic extracts (10 μg/mL and 1 mg/mL) on xanthine/xanthine oxidase (X/XO)-, FeCl2/ascorbicFeCl2/ascorbic acid- and arachidonic acid (AA)- induced neurotoxicity in mouse cortical cell cultures.	The highest concentration showed a reduction of 89 and 100% of the toxic effect exerted by X/XO and FeCl2/ascorbicFeCl2/ascorbic acid, respectively. While the neuronal injury induced by AA decreased by 22%. Presumably, OFI exerts protection against certain neuronal injuries caused by the excessive presence of free radicals.	[80]
In vitro	Quercetin, (+)-dihydroquercetin, and quercetin 3-methyl ether were isolated from EtOAc fractions originating from OFIS fruits and stems to determine their protective effect against H_2_O_2_ and X/XO induced neuronal injury in primary cultured rat cortical cells.	Results indicate that all flavonoids decreased neuronal injury and significantly inhibited lipid peroxidation. However, quercetin had the best effect, with an IC_50_ between 4 and 5 μg/mL. It is suggested that these active ingredients have a neuroprotective action related to their antioxidant capacity.	[81]
In vitro In vivo	In the first study, a methanolic extract of OFI (MEOFI) was tested against neuronal injury induced by NMDA, KA and oxygen-glucose deprivation (OGD) in cortical cell culture from mouse. Subsequently, it was evaluated its protective effect in the CA1 region of the hippocampus against neuronal damage caused by global ischemia in gerbils.	The treatment of the extract (30, 300 and 1000 μg/mL) inhibited the neurotoxicity induced by NMDA, KA and OGD in a dose-dependent manner in cortical cells. Likewise, in animals previously treated with MEOFI (0.1, 1.0 and 4.0 g/kg, p.o.) every 24 h for 3 days and 4 weeks, the neuronal damage in the hippocampus was reduced by approximately 35%;suggesting that preventive administration of MEOFI can alleviate excitotoxic damage induced by global ischemia.	[82]
In vitro	Considering that high concentrations of ROS, especially superoxide anion (O_2_^-^) and peroxynitrite (ONOO^-^), product of the NO reaction, contribute to the oxidative toxicity generated in NDs. The neuroprotective activity of two butanolic fractions prepared from the 50% ethanolic extract of OFIS stems was evaluated.	Both the stem fraction (SK OFB901) and its hydrolysis product (SK OFB901H) inhibited the NO production in microglia activated by LPS in a dose-dependent manner. In addition, they suppressed iNOS mRNA expression in microglia cells observed by western blot analysis and RT-PCR. These results demonstrate that the neuroprotective property of OFI is through the reduction of NO by activated microglial cells and the uptake of ONOO^-^	[83]
In vitro	The results obtained by Lee et al., (2006) [83] motivated another experiment with the SK OFB901 fraction to determine its action on neuronal lesions induced by EOx, excitotoxins and Aβ in primary cultured rat cortical cells. In addition, through cell-free bioassays, its antioxidant potential was determined.	SK OFB901 inhibited H_2_O_2_- and X/XO-induced neuronal damage and Glu-, NMDA- and KA-induced excitotoxicity. Likewise, the neurotoxicity exerted by Aβ and the lipid peroxidation initiated by Fe^2+^ and L-ascorbic acid in rat brain homogenates were attenuated. All these data indicate that the butanolic fraction has antioxidant and neuroprotective capacities through multiple mechanisms, which implythe possible application to prevent and treat NDs.	[84]
In vitro In vivo	Huang et al., (2008, 2009) carried out two experiments in order to determine the neuroprotective effects of Cactus polysaccharides (CP) extracted from OdHw. In the first, this capacity was evaluated on the damage induced by OGD and reoxygenation (REO) in the cortical and hippocampal slices of rat brain.Cell viability and quantification of cell survival were quantified using the 2, 3, 5-triphenyl tetrazolium chloride (TTC) method and The fluorescence of propidium iodide (PI) staining, respectively. Subsequently, they analyzed the mechanisms of ischemia-reperfusion injury of the middle cerebral artery in Sprague-Dawley rats and the damage induced by EOx in PC12 cells.	Both studies demonstrated that: (a) The ischemic condition decreased cell viability and increased lactate dehydrogenase (LDH) release, (b) CP protected brain slices from the OGD injury, decreased PI intensity and LDH release. Likewise, it prevented the increase in iNOS activity, (c) With a dose of 200 mg/kg of CP, the volume of the infarction and the neuronal loss in the cerebral cortex of rats were reduced, and d) Finally, the in vitro conditions confirmed that the CP pretreatment significantly increases cell viability, protects PC12 cells from H_2_O_2_ damage, and reduces apoptosis and ROS production. The results suggest that the protective mechanism of CP may be partially mediated by the NO/iNOS system and induced by the OGD aggression. In addition, it can be considered a candidate compound for the treatment of ischemia and DN induced by EOx	[85] [86]
In vitro	The objective was to determine the chemical constituent of *O. Milpa Alta* polysaccharides (MAP) and its neuroprotective potential in an in vitro model of cerebral ischemic injury. By using the gas chromatograph and GC-MS it was observed that MAP mainly contained galactose, arabinose, rhamnose and glucose.	On the other hand, the three concentrations of MAP (0.5, 5, and 50 μg/mL) increased the cell viability [methylthiazolyltetrazolium (MTT) assay], inhibited LDH-induced cellular cytotoxicity, suppressed ROS production and decreased intracellular Ca^2+^ concentrations; significantly preventing neuronal cell death.	[87]
In vitro	Inflammation associated with microglia activation in neuronal injury can be achieved by exposure to LPS. Thus, using 4 different serotypes of LPS, a differential effect related to β-oxidation of fatty acids and antioxidant enzymes in peroxisomes was identified.	Using various OFI cladode extracts, an antioxidant effect was demonstrated through microglial catalase and an anti-inflammatory effect by reducing the production of NO LPS-dependent; suggesting that these extracts have a neuroprotective activity through the induction of peroxisomal antioxidant activity.	[61]
In vitro	A hallmark of age-related neurodegenerative proteinopathies is the misfolding and aggregation of proteins, usually Aβ in AD and α-synuclein (α-syn) in PD, which in soluble oligomeric structures are often highly neurotoxic. Using two different experimental models, we investigated whether prickly pear extracts from OFI (PPEOFI) alleviated the neurodegenerative effects of AD and PD in yeast (*Saccharomyces cerevisiae*) and fly (*Drosophila melanogaster*).	Pretreatment with PPEOFI in the culture medium increased the viability of yeast expressing the Arctic mutant Aβ42 (E22G). Likewise, dietary supplementation of PPEOFI dramatically improved the lifespan and behavioral signs of flies with brain-specific expression of wild-type Aβ42 (late-onset AD model) or the Arctic variant of Aβ42 (early-onset AD model). Increased fly survival was observed in a PD model where the human α-syn A53T mutant is expressed. These findings indicate that PPEOFI interferes with the neurodegenerative mechanisms of AD and PD. Probably they inhibit both Aβ42 and α-syn fibrillogenesis by accumulating remodeled oligomeric aggregates that are less toxic to the lipid membrane.	[88]
In vitro	The purpose of the study was to detect the specific areas of the brain where Ind, derived from OFI, can be localized in significant quantities after an oral administration and to highlight its possible local effects on the excitability of individual neuronal units.	HPLC analysis of brain tissue after ingestion of 2 μmol/kgInd indicates that it accumulates primarily in the cortex, hippocampus, diencephalon, brainstem, and cerebellum. Using electrophysiological recordings and microiontophoretic technique, its influence on neuronal firing rate was evaluated, confirming that neuronal bioelectrical activity is modulated after the local injection of Ind. These findings constitute the justification to explore the biological mechanisms through which bioactive compounds could modulate the neuronal function with a relapse in the cognitive brain process and neurodegenerative conditions.	[89]
In vitro	Different studies agree that Ind has anti-inflammatory and neuromodulatory effects. Therefore, discovering new physiological targets plays an important role in understanding its biochemical mechanism. In this regard, combined reverse pharmacophore mapping, reverse docking, and a search of some databases identified Inositol Trisphosphate 3-Kinase, Glutamate carboxypeptidase II, Leukotriene-A4 hydrolase, Phosphoserine phosphatase, Phosphodiesterase 4D, and Kainate receptor (GluK1 isoform) as potential targets for indicaxanthin.	The results suggest that these targets are involved in neuromodulation and inflammatory regulation, normally expressed in the CNS and in cancerous tissues (especially breast, thyroid and prostate). Furthermore, this study provides insights into the dynamic interactions of Ind at the binding site of target proteins, through molecular dynamics simulations and MM-GBSA.	[90]
In vitro	Although several studies have reported that OFIS has antidiabetic, antiasthmatic and analgesic properties; its action mode is not clearly described. Therefore, the anti-inflammatory and neuroprotective capacity of an ethanolic extract (EEOFIS) was analyzed individually or in combination with Vitamin C (Vit C). NO, iNOS and COX-2 levels were evaluated in macrophage cells. In addition, a cell viability assay was performed to confirm its protection against Aβ-induced neurotoxicity.	At the conclusion of the study, elevated levels of cAMP response element-binding protein were found and brain-derived neurotrophic factor expression upon evaluation of the regulation of synaptic plasticity by EEOFIS in SH-SY5Y neuroblastoma cells. A synergistic effect was also confirmed in the combined treatment with Vit C. These results suggest that EEOFIS can improve the cognitive function through an anti-inflammatory response, cell protection and regulation of synaptic plasticity.	[91]
In vivo	In this work, 37 OFI metabolites were characterized using HPLC-MS/MS and the main polysaccharides of its fruit pulp and CLDs were identified, as well as their neuroprotective activity under in vitro conditions of AD induced by AlCl3	All the tested extracts presented antioxidant activity; however, the most representative effects were for those from CDLs (possibly due to their high phenolic content). A significant decrease in learning and memory impairment induced by AlCl3 was observed (Passive avoidance test). In addition, elevated brain levels of proinflammatory cytokines (NF-κB and TNF-α) were reduced.	[92]
In vitro	Organic extracts of spines, flowers, roots and fruits of *O. microdasys* var. rufida (OMR) and *O. leptocaulis* (OL) were studied for their phytochemical composition and their anticholinesterase, cytotoxic and neuroprotective activity. The catalase test result was that the extracts have a potent antioxidant activity. The anticholinesterase activity was determined by butyrylcholinesterase (BChE) and revealed that all extracts were endowed with excellent inhibitory efficacy against BChE; however, the EtOAc extract of OMR flowers was the most significant	On the other hand, the neuroprotective effect of the extracts was evaluated against the toxicity induced by Aβ in PC12 cell lines; confirming that the MeOH extract of OMR spines was the one that increased cell viability the most (approximately 80%). The MTT assay showed that the extracts presented an evident cytotoxic activity on HeLa cells. Finally, the column chromatography of the EtOAc extract of OMR flowers identified 5 flavonol glycosides (isorhamnetin-3-O-α-rhamnopyranosyl-(1 → 2)[α-rhamnopyranosyl-(1 → 6)]-β galactopyranoside, quercetin-3-O-β-pyranogalactoside (hyperoside, isorhamnetin-3-O-β-galacto (1 → 6)-α-rhamnoside, isorhamnetin-3-O-β-glucoside and kaempferol-3-O-β-arabinoside	[93]

**Table 5 plants-11-02333-t005:** Scientific evidence of the antimicrobial effects of *Opuntia* spp.

Type of Study	Objective and Characteristics	Results and Conclusion	Ref
In vitro In vivo	The food industry is continually looking for ingredients that provide advantageous properties to food products, especially protecting their nutritional value. Because crude oils are examples of this type of ingredient, the in vitro and in vivo bioactivities of *O. macrorhiza* Engelm (OMESO) seed oil were chemically characterized and evaluated.	OMESO presented a low acidity index, oxidation stability and a high content of unsaturated fatty acids. It also showed antioxidant activity, cytotoxicity against human tumor cell lines and antibacterial capacity, especially against Gram (+) species such as *S. aureus*, *E. faecalis*, *B. cereus* and *L. monocytogenes*. The latter was the one with the greatest diameter of inhibition. These properties and its low toxicity in animals favor the use of OMESO compared to synthetic bioactive agents (ampicillin, amphotericin B) that induce greater adverse effects.	[60]
In vitro	The aim of the study was to determine the antimicrobial effects of Mexican medicinal plant extracts, including OX, against some pathogenic bacteria [both Gram (+) and Gram (−) species] using the disk diffusion assay. The cytotoxicity of the extracts on human breast cancer cells (MCF-7) was also evaluated with the MTT assay.	Most of the extracts evidenced this beneficial effect. However, OX presented the best antimicrobial capacity against *Acinetobacter lwoffii* and more resistant strains such as *A. baumannii*, *S. aureus* and *S. haemolyticus*. Although, unfortunately, it did not show any cytotoxic action on MCF-7 cells; unlike the extract of *Justicia spicigera* and *Phoradendron serotinum*.	[107]
In vitro	Adherence and cytotoxicity of *Campylobacter* spp to host mucosa are critical steps in inducing bacterial gastroenteritis. The proposal is to use natural food products to reduce its pathogenesis. With that purpose, the bactericidal potential of 28 plant species (including OFI) on the growth of *C. jejuni* and *C. coli* was analyzed.	The OFI extract was one of the most effective against both microorganisms at MBCs of 0.3, 0.5, 0.4 and 2.0 mg/mL. This same extract also diminished the adherence and cytotoxicity of bacteria on Vero cells. Thus, OFI may be a candidate for the control of food contamination by Campylobacter and/or as a feed supplement to reduce the prevalence of this bacterium on farms.	[108]
In vitro	In this investigation, the antimicrobial activity of an extract of PPFs from OX against four strains of *Escherichia coli* O157: H7 was determined, by means of brain-heart infusion (BHI) medium to analyze bacterial growth over time and in agar well diffusion.	The results showed that the extract had a significant inhibitory effect at concentrations of 4.0, 6.0, 8.0 and 10% at 8 h of incubation at 37 °C which confirmed that such potential was concentration dependent. Therefore, prickly pear fruits of OX could be considered a natural means to control the pathogenic contamination in food and reduce its risks.	[109]
In vitro	The chemical composition of hexane extracts from flowers belonging to two species of *Opuntia* (OFI and OdHw) were studied by gas chromatography–mass spectrometry in four developmental stages of flower (vegetative, initial flowering, full flowering, and post-flowering stages).	The differences observed in the composition ofthe two species of flowers were mainly carboxylic acid, terpenes, esters and alcohols. Furthermore, both *Opuntia* species showed inhibitory activity against *P. aeruginosa*, *S. aureus* and *E. coli*. Therefore, OFI and OdHw could be used as food preservative agents.	[110]
In vitro	Because there are few reports on the chemical composition and biological activity of OFI in its flowering development, the percentage of nutrients and antibacterial activity of a hexane extract of its flowers were studied in different 4 stages of flowering.	The results showed that during flowering there were no significant variations in its chemical composition; finding you mainly fiber, proteins and minerals. Also observed a high efficacy against *E. coli* and *S. aureus*, making it a botanical source with possible additive food control potential.	[111]
In vitro	Some evidence suggest that OdHw endophytic fungi may help the host to overcome biotic and abiotic stress by producing biologically active metabolites. To confirm this, we evaluated the antimicrobial activities of endophytes isolated from their CLDs and flowers against 5 bacteria [3 Gram (+) and 2 Gram (-)].	Of the 8 fungi isolated, *Fusarium* spp was the most bioactive and presented equisetin (derived from tetramic acid) as an antibacterial compound. Their MBCs ranged from 8 to 16 μg/mL for *S. aureus* and Methicillin Resistant *S. aureus* (MRSA). The conclusion is that these fungi can help the host to resist the stressful environmental conditions and produce biologically active secondary metabolites.	[112]
In vivo In vitro	Ammar et al., (2015) investigated the healing (excision wound model in rats), antioxidant (Trolox equivalent antioxidant capacity and DPPH assay) and antibacterial (agar-well diffusion assay) activity of mucilaginous and methanolic extracts from OFI flowers.	The conclusión was that both extracts showed significant results and the mucilage extract was the most effective. In practically all the microorganisms tested (*L. monocytogenes*, *E. coli*, *P. aeruginosa*, *S. aureus*, and *B. subtilis*) inhibition halos were observed. However, *L. monocytogenes* was the most sensitive	[113]
In vitro	Initially, the synthesis of HAP nanoparticles was made using pectin (extracted from the shell of prickly pear fruits from OFI) as a base template. Subsequently, the evaluation focused on its efficiency and antimicrobial activity against *S. aureus* and *E. coli* in the absence and presence of pectin at a concentration of 0.15%.	The results showed that the HAP nanoparticles synthesized with pectin had better antimicrobial activity against both bacteria compared to those without pectin. On average, the inhibition halos ranged from 6 to 8 mm in diameter. Therefore, these little crystalline and granular nanoparticles can be useful in the field of biomedicine.	[114]
In vitro	Biofilm is a complex microbial community that is highly resistant to antimicrobial agents, and its formation is associated with high rates of morbidity and mortality in hospitalized patients. Considering that the use of medicinal plants is a new proposal for the control of hospital infections, the antimicrobial and antibiofilm activities of 8 methanolic plant extracts were evaluated, (including OFI) against 5 nosocomial pathogens (*K. pneumoniae*, *E. faecalis*, *E. coli*, *S. maltophilia* and *S. aureus*).	Preliminary antimicrobial tests performed by the well diffusion method showed that OFI induces zones of inhibition ranging from 0.7 to 1.3 cm and the MBCs were between 1.0 and 15 mg/mL. Besides, most pathogens were inhibited and the most sensitive were *E. coli* and *S. aureus*. The specific biofilm formation index (SBF) was evaluated before and after the addition of plant extracts and again OFI caused the greatest reduction in SBF.	[115]
In vitro	The presence of multiresistant pathogenic bacteria in food is known to be a major public health problem, especially Diarrheagenic *Escherichia coli* pathotypes (DEPs). In the case of nopalitos (raw whole and chopped) and in nopalitos salad samples, generic E. coli and multiresistant DEPs were found. The generic *E. coli* was determined using the most probable number procedures and for DEPs two multiplex polymerase chain reaction procedures were used and their susceptibility to 16 antibiotics was evaluated for the DEPs strains.	Of the 300 samples of nopalitos (100 for each type evaluated), both generic *E. coli* and DEPs between 10 and 80% per type were identified in them. The DEPs that were identified, include Shiga toxin-producing *E. coli*, enteropathogenic *E. coli*, and enterotoxigenic *E. coli*. Finally, all the isolated strains exhibited resistance to at least six antibiotics.	[116]
In vitro	The objective of the study was to characterize the phytochemical profile and determine the cytotoxic and antimicrobial properties in flowers of *O. microdasys* (OMs) at different stages of maturity. An initial observation was that OMs stand out for their high content of dietary fiber, potassium and camphor.	The vegetative stage showed the highest cytotoxic and antifungal (*A. versicolor* and *P. funiculosum*) activities, while the full bloom stage was particularly active against bacterial species (*S. aureus*, *B. cereus*, *M. flavus*, *L. monocytogenes*, *E. coli*, *P. aeruginosa*, *S. typhimurium*, and *E. cloacae*). Of these, *S. aureus* was the most susceptible species, while *L. monocytogenes* and *E. cloacae* stood out as the most resistant.	[117]
In vitro	Seed oils extracted with different solvents (Hx, EtOH and EtOAc) and from two Mexican varieties of PPFs [red: OFI and green: *O. albicarpa* (OA)] were evaluated to determine their antioxidant and antimicrobial activity. The fatty acid profile of the oils was also quantified by gas chromatography-mass spectrometry (GC-MS), which confirmed that both varieties of PPFs were similar and exhibited a high content of linoleic acid.	Because OA oil obtained with EtOH and EtOAc showed the highest antioxidant activity (323 and 316 μmol TE/20 mg, respectively), it was used to analyze the antimicrobial potential; which showed inhibition halos in most of the microorganisms evaluated (*E. coli* O58:H21 and O157:H7, *S. aureus*, *L. monocytogenes*, *P. aeruginosa*, and *S. Typhi*). *S. typhi* and *E. coli* O157:H7 were the most resistant species.	[118]
In vitro	Various evidences have confirmed that the oils from the seeds of *Opuntia* species have a significant content of unsaturated fatty acids and antioxidant compounds. Therefore, the focus of the study was to compare the effectiveness of conventional extraction methods (extraction with hexane) and new ones (supercritical (SC)-CO_2_) for oil recovery, obtaining phenolic compounds and action of the antimicrobial effect of the *O. stricta* seeds.	Using liquid chromatography-high-resolution mass spectrometry, the conclusion is that similar yields of oil are obtained in both extraction methods; although when using the SC-CO2 method, it is more enriched in polyphenols, which favors an increase in antioxidant potential and its percentage of antibacterial inhibition (especially against *B. thuringiensis* and *B. subtilis*). This extraction method favors the beneficial properties of *O. stricta* to suggest its oil as high quality.	[119]
In vitro	Aqueous extracts of OFI, *Artemisia herba-alba*, *Camellia sinensis* and *Phlomis crinita* were evaluated by the disc method against two Gram-negative bacterial strains (*Porphyromonas gingivalis* and *Prevotella intermedia*) commonly involved in periodontal diseases.	All extracts showed a powerful activity against these strains, especially OFI whose inhibitory concentration varied between 0.03 and 590 mg/mL. In summary, the statistical analysis showed that the most significant antimicrobial effect was on *P. intermedia*	[120]
In vitro	Considering the previous results of Gómez-Aldapa et al. (2016) [105], in this study the presence of Salmonella strains resistant to antibiotics in nopalitos (raw whole and cut) and in samples of nopalitos salad was found. The analysis also covered the behavior of multiresistant Salmonella isolates.	Bacterial strains were found between 10 and 30% of the samples. From all the samples, 70 multiresistant Salmonella strains were isolated, which survived longer in whole raw nopales at 25 °C; unlike the strains found in the other nopalitos samples where their growth was inhibited at 3 °C. Possibly, this is an important factor that contributes to alimentary gastroenteritis.	[121]
In vitro	Antimicrobial resistance is a serious health problem of the 21st century, which is intended to be solved by searching for new agents with this therapeutic property in plants. Consequently, fresh OFI fruits were collected to extract their bioactive compounds using solvents such as EtOH, MeOH, Chl. Afterwards, the antimicrobial potential of these extracts against *E. coli*, *S. pneumoniae*, *S. typhi* and *B. subtilis* was determined by the diffusion method in agar wells.	All the extracts demonstrated antibacterial activity against the 4 bacteria, showing an inhibition diameter between 9.0 and 23.0 mm. The highest activity was against *S. typhi*, *B. subtilis* and *S. pneumoniae*, whch was significantly higher when compared to synthetic antimicrobials (tetracycline and vancomycin). These results suggest that OFI extracts could be used for prevention and treatment of different bacterial diseases.	[122]
In vitro	With the purpose of expanding the knowledge about natural bioactive compounds for food preservation, an aqueous extract of purple-red prickly pears obtained from the first flowering of OFI and the total content of polyphenols, betacyanins, and betaxanthins was evaluated; as well as the antimicrobial against food spoilage induced by different pathogenic bacteria (*E. coli*, *S. enterica*, *P. fluorescens*, *L. innocua*, *S. aureus*, *B. subtilis*, *B. cereus*). The extract was applied through the immersion technique to sliced beef meat in order to determine its physical and chemical parameters, and maintenance of color and texture.	The addition of the extract preserved the color, texture and extended the shelf life of the meat during the storage period. Likewise, the agar well diffusion test showed that the extract has a broad-spectrum activity by inhibiting the growth of all bacterial strains; especially against *B. cereus*, *P. fluorescens* and *E. coli*. These results support the possibility that the betacyanins and betaxanthins of the extract favor the general quality of the meat under refrigeration conditions.	[123]
In vitro	Considering the previous results of Ammar et al., (2015) [102], the antimicrobial, antifungal activity and skin wound healing effect of the oil extracted from the seeds of PPFs from OFI were evaluated. For the first properties, minimal inhibitory concentrations (MICs) and minimal bactericidal concentrations (MBCs) were calculated against 4 bacterial strains (*E. coli*, *S. aureus*, *S. agalactiae*, and *E. cloacae*), and 3 fungi (*A. niger*, *P. digitatum*, and *F. oxysporum*).	The oil was able to mainly inhibit *E. cloacae*, *A. niger*, *P. digitatum* and *F. oxysporum*. (On average, the inhibition halos were 16 mm compared to the 23 mm obtained by the positive control of Ceftazidime). In addition, a good wound healing effect was observed, preventing skin infections and reducing the re-epithelialization phase. These data suggest that OFI oil exerts both bacteriostatic and bactericidal effects on *E. cloacae* and appears to be effective for the treatment of skin infections.	[124]
In vitro	Sexually transmitted infections (STIs) continue being a major health problem and unfortunately, antimicrobial drugs are becoming ineffective due to the increasing resistance of bacteria and viruses; thus, the use of medicinal plants has become a good alternative. Using the disk diffusion model and the microdilution technique to determine the zone of inhibition and MICs, some plant extracts (including OFI) were tested against *N. gonorrhoeae* and some fungal strains (*C. albicans*, *C. krusei*, *C. parapsilosis*, *C. tropicalis* and *C. neoformans*)	The extracts (MeOH, Hx, Ace and DCM) presented different levels of phytoconstituents such as alkaloids, steroids, terpenes, flavonoids, tannins and saponins. Especially the Ace and MeOH extracts of OFI showed potency against *N. gonorrhoeae* and fungal strains. These results open the field of new studies to consider plants as an alternative method for STIs control.	[125]
In vitro	The shell of the PPFs is usually an agroindustrial waste that has been little studied in the nutraceutical area. Consequently, the main components of the shell were isolated and characterized in order to subsequently quantify their antibacterial capacity. Initially, a MeOH extract was fractionated using Hx, Chl and EtOAc. The GC-MS analysis confirmed that the Hx fraction had 60% linolenic acid; while the study of the EtOAc fraction by ultra-performance liquid chromatography electrospray tandem mass spectrometry (UPLC-ESI-MS/MS), revealed caffeic acid and quercetin.	This EtOAc fraction was also subjected to column chromatography, resulting in the isolation of four flavanols (astragalin, quercetin 5,4′-dimethyl ether, isorhamnetin-3-O-glucoside, and isorhamnetin). The antibacterial evaluation revealed that the EtOAc fraction (specifically, quercetin 5,4′-dimethyl ether) was more potent against pneumonia pathogens. These findings indicate that OFI fruit debris containvaluable components against some pathogens.	[126]
Systematic review	Of all the studies carried out to date, this is the only one where a bibliographic search has been carried out on fruit extracts and agro-industrial residues with antimicrobial activity that can be applied to meat products.	The data obtained confirm that: a) *Opuntia* extracts have antimicrobial effects against *L. monocytogenes*, *B. cereus*, *S. aureus* and *E. coli*, b) Other important extracts and/or by-products were those of the grape that show inhibition of *S aureus*, *L. monocytogenes*, *P. aeruginosa*, *E. coli*. These data reinforce the possibility of substituting synthetic preservatives by natural versions. For this reason, it is necessary to investigate in detail the effective concentrations that maintain the sensory properties of foods.	[127]
In vitro	By comparing two fractionation processes (semi-preparative high-performance countercurrent chromatography (HPCCC) and HPLC) of *O. stricta* extracts to obtain secondary metabolites, it was confirmed that HPCCC has a better separation capacity; obtaining two 14-ring cyclopeptide alkaloids (Opuntisine A and B)	In determining its antimicrobial potentials, we found that opuntisin A had moderate activity against *E. coli*. Therefore, a strong suggestion is to extend the studies in these new natural products of the Cactaceae family.	[128]
In vitro	Urinary tract infections (UTI) are caused by different microorganisms, highlighting *E. coli* in 90% of female cases. Considering all previous OFI studies, the antimicrobial efficacy of ethanolic and ethyl acetate extracts of the OFI cactus on this Gram (−) bacterium isolated from UTI patient samples was assesed. The results were compared against reference antibiotics (gentamicin and ampicillin).	An important observation was that the EtOH extract had a higher activity against *E. coli* compared to EtOAc and the reference antibiotics. In the same way, it was established that the EtOAc extract may have activity on bacteria from food (*B. subtilis*, *S. aureus*, *S. typhimurium* and *P. fluorescens*). In conclusion, the inhibitory effect of both extracts against Gram (+) and (−) bacteria can be attributed to the presence of the different bioactive ingredients.	[129]

**Table 6 plants-11-02333-t006:** Scientific evidence of *Opuntia* spp. on the healing of skin wounds.

Type of Study	Objective and Characteristics	Results and Conclusion	Ref
In vitro	As mentioned, OFI flowers are used for various medicinal purposes. Therefore, the healing activity (excision wound model in rats) and antioxidant activity (Trolox equivalent antioxidant capacity and DPPH assay) of mucilaginous and methanolic extracts of its flowers were studied.	After 13 days of treatment with both extracts, a beneficial effect on skin repair was observed, evaluated by the acceleration of the phases of contraction and remodeling of the wound. Histopathological studies of the granulation tissue indicated that the dermis was properly corrected and that the mucilage extract was more effective. In addition, it was confirmed that the extracts showed a significant antioxidant capacity.	[113]
In vivo	After isolating, washing, drying and cold pressing PPFs seeds from OFI, their oil was obtained to determine the effect of cicatrization of skin wounds and its antimicrobial potential against 4 bacterial strains and 3 fungi. The skin wounds of three experimental groups of rats were topically treated once a day with the oil, observing the healing process and calculating the percentage of wound contraction. At the same time, a histological study was performed on skin biopsies.	At the end of the study, it was shown that the oil exerted a good wound-cicatrization effect, preventing skin infections (especially against *E. cloacae*, and *A. niger*) and reducing the re-epithelialization phase. It is suggested to increase the studies to confirm the capacity of the oil in the promotion of the cicatrization process.	[124]
In vivo	The purpose of this study was to investigate the effect of spraying an extract of *O. stricta* on wounds on the ventral surface of rabbit ears. After the wounds healed, hypertrophic scar tissue was obtained and histological analysis was performed. Using immunohistochemistry and real-time quantitative polymerase chain reaction, the expression of type I and III collagen and matrix metalloproteinase-1 (MMP-1) was evaluated.	The results indicated: a) the expression of type I collagen in the animals treated with the extract was lower than in the control group, unlike type III collagen that gradually increased, b) the scar that was less prominent and expression of MMP- 1 decreased with the application of the extract. In conclusion, the extract decreased the formation of hypertrophic scars by inhibiting type I collagen, and increasing type III collagen and MMP-1.	[138]
In vivo	Despite advances in modern medicine, to date there is no effective natural treatment for second-degree burns. Therefore, the healing efficacy of oil extracted from PPFs on partial-thickness burns induced by fractional CO2 laser in rats was evaluated. All the burns were measured and treated topically for 7 days. The response to treatment was determined by macroscopic, histological and biochemical parameters.	The oil showed improvements in the general appearance of the wound and in the formation of scabs; besides, it significantly decreased the healing time. The histological evaluation confirmed that the oil has comparatively good healing properties and favors collagen content. This is scientific evidence of the efficacy of PPF oils on partial thickness burns.	[139]
In vitro	The purpose of the research was to compare the effects of OFI and Milk Thistle (MT) (*Silybum marianum* L.) on adult keratinocytes (HaCaT) functioning in basal conditions or in the presence of mechanical damage (wounded cells). Natural compounds were tested on HaCaT in monoculture and triculture configurations. In three-culture models, HaCaTs were treated with conditioned media obtained by co-cultures of normal human dermal fibroblasts and human dermal microvascular endothelial cells.	After determining cell viability, mechanisms of EOx (cytokine release and lipid peroxidation), cell remodeling (modulation of metalloproteinases), and migratory potential of HaCaT (in vitro wound healing assay); OFI and MT were found to favor migratory properties of HaCaT under both physiological conditions and mechanical damage. In addition, the response to EOx was modulated. The conclusion was that OFI and MT are good alternatives in skin repair.	[140]
In vivo	This last study investigated the potential of opuntiol, isolated from OFI, against UVA radiation-mediated inflammation and skin photoaging in mice. The animals were shaved and exposed to UVA rays (dose of 10 J/cm^2^/day) for ten days. One hour before each exposure, opuntiol (50 mg/kg) was applied topically.	Opuntiol pretreatment prevented UVA-linked clinical macroscopic skin lesions and histological changes in the mouse skin. In addition, opuntiol prevented dermal collagen fiber loss and collagen I and III breakdown in animal skin. Opuntiol was found to inhibit UVA-induced activation of iNOS, TNF-α, COX-2 MMP-2, and MMP-9. In conclusion, opuntiol exerted skin protection to the photoaging response associated with UVA radiation by reducing inflammatory responses and activating MAPK.	[141]
In vitro	*O. humifusa* (OHF) is considered a possible candidate to design cosmetic formulations that prevent the harmful effects of Particulate Matter (PM). Unfortunately, its high viscosity does not allow its adequate use in these formulations. Therefore, the effect of a high-power microwave treatment on an *O. humifusa* extract (MA-OHF) was investigated.	The results indicated that MA-OHE showed reasonable viscosity and outstanding anti-inflammatory activity to suppress PM-induced ROS production. In addition, COX-2 and MMP-9 expression was decreased in HaCaT keratinocytes. It is suggested that MA-OHE may be a suitable natural cosmetic ingredient to prevent PM-induced skin oxidative stress and inflammation.	[142]
In vivo	Considering that delayed wound healing represents a common health hazard, we compared the wound cicatrization activity of OFI seed oil and an auto-nanoemulsifying drug delivery system (OFI-SNEDDS) formulation in a full-thickness skin excision rat model. The OFI-SNEDDS formulation was prepared using a droplet size of 50.02 nm and applied directly to the animals.	The results showed that the formula exhibited healing activities superior to the oil, which was confirmed by histopathological examinations. In addition, OFI-SNEDDS presented greater antioxidant and anti-inflammatory capacity and improved angiogenesis (a phenomenon that was demonstrated by increasing the expression of vascular endothelial growth factor). The conclusion was that OFI has wound healing properties that are enhanced by the self-emulsion of the oil in nanodroplets. This is probably attributed to its anti-inflammatory, procollagenous and angiogenic properties.	[143]
In vitro	In this study, chitosan-based wound dressings loaded with an OFI extract were prepared. Chitosan (Ch) was crosslinked with a low molecular weight diepoxy-poly(ethylene glycol) (PEG), and hydrogel films with different Ch/PEG composition and OFI content were prepared. Using FTIR spectroscopy (Fourier transform infrared spectroscopy) the appearance of the crosslinking reaction was determined.	The analyses suggested that ionic interactions between Ch and OFI occur. The swelling characteristics, the water vapor transmission rate and the release kinetics showed that these films are suitable for their application. Finally, a scratch test on a keratinocyte monolayer showed that the rate of cell migration in the presence of OFI-loaded samples is approximately 3 times higher compared to unloaded films, confirming its restorative activity.	[144]

## Data Availability

Not applicable.

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
