# Peer review of "Opuntia spp. in Human Health: A Comprehensive Summary on Its Pharmacological, Therapeutic and Preventive Properties. Part 2"

_plants, 2022, doi:10.3390/plants11182333_

Round 1

Reviewer 1 Report

Title;  Comparative analyses and molecular videography of MD simulations on WT human SOD1
Comments; In my view, the results showed in this review are worthy for publication. The manuscript needs major essential revision before publication. I would like to overview the revised version of the manuscript before it accept for publication. I have the following comments/suggestions for authors to address before final decision on the manuscript.
1. Re-frame the heading of section 3.
2. “It should provide a concise and precise descriptionOpuntia spp is a diverse and widely distributed genus in the American Continent.”: Incomplete sentence/missing punctuations.
3. Show all the five species (O. streptacantha (OS), O. hyptiacantha (OH), O. albicarpa (OA), O. megacantha (OM) and O. ficus-indica) mentioned in the text in Figure 1.
4. Mention in a few sentences the reason for “only 1.5% of this production is exported [4,7,11,12].”
5. Add more Figures in the manuscript to make the data more appealing to a general audience.
6. In the Introduction section the author should refer to the research paper and comment on recent in-silico techniques on plant Bioactive.  It will be good information for the readers. I would like to recommend several papers, among many others, providing further explanation on this topic:  PMID: 21989830 PMID: 23383724 PMID: 35604288 PMID: 35362492 PMID: 35276295 PMID: 35315127 PMID: 3548651
7. Authors have suggested adding details about Opuntia found in other regions/countries.
8. Authors have to elaborate on the therapeutic values of Opuntia and its role described in different studies.
9. Authors have advised to add and compare the theoretical data about Opuntia in the manuscript.
10. Authors have to add some graphical data regarding Opuntia in the manuscript.
11. Authors have mention a lot of studies on different pharmacological effects of Opuntia spp. I thought the authors should find a way to incorporate figures in the review article.
12. In case of some diseases if exact mechanism is not known then authors should depict the possible mechanism with the help of figures.
13. At a movement it looks major drawback of this review article. It looks very descriptive. For e.g section “4.4. Action in the Treatment of Skin Wounds” could be better represented with the help of suitable figure.

Author Response

Dear reviewer

The authors appreciate the comments and observations of the article

We have considered all suggestions and observations

Please check the attached file

Thanks for everything

Receive a cordial greeting

We believe that there is a confusion with the title mentioned “Title; Comparative analyses and molecular videography of MD simulations on WT human SOD1”.

There is no relation to the topic of our document

  1. We have considered the suggestion. The subtitle of section 3 was re-formulated
  2. We have considered the suggestion. We changed the subtitle of section 2 (“Impact of the Opuntia genus in Mexico and other countries”). A brief paragraph was included that adequately described the Cactaceae family and the Opuntia genus.

Within the text, a suggestion was made “This section may be divided by subheadings. It should provide a concise and precise description”. We do not believe that it is appropriate to divide the section, since the topic is very broad and are only trying to comment on general and introductory aspects.

  1. We have considered the suggestion. The 5 species were included in figure 1.
  2. We have considered the suggestion. A brief paragraph was included where the factors that affect this percentage are mentioned.
  3. We appreciate the suggestion, but we believe that including more figures would increase the size of the document and change the general structure. However, we have included a figure on wound healing and two more tables (Table 1 and 2)

Table 1. Main products and by-products obtained from Opuntia (Nopal)

Table 2.Nutritional composition in different anatomical parts of Opuntiaficus-indica (L.)Mill.

  1. We have considered the suggestion. A paragraph was included in the conclusions and perspectives section. The relevance of in silico studies and opuntia evidence in this area of research was mentioned.

We appreciate the suggested references, but include those of opuntia studies. This topic is relevant and extensive, so it could be another review

  1. We have considered the suggestion. We changed the subtitle of section 2 (“Impact of the Opuntia genus in Mexico and other countries”). A brief paragraph was included to describe the impact (positive and negative) of opuntia in Mexico and other countries.
  2. Speaking or specifically describing therapeutic values of opuntia is complicated, since its species are frequently used as food, traditional medicine and in the cosmetic industry

Most studies agree that Opuntia is safe.

However, we include a brief paragraph in the Toxic Evidence of the genus Opuntia section where reference is made to these possible therapeutic values and the dose and/or concentration ranges that have been mentioned in both documents (part 1 and part 2)

  1. We value the comment and observation, but unfortunately we are confused.

We believe that the information (theoretical and/or practical studies) is being described and the comparison or analysis is considered. We honestly don't understand the suggestion, and it doesn't match the other reviewer's comments.

  1. We believe that the structure of the document makes an analysis, comparison and description of the results in each study.

We do not believe that it is essential to include graphic data. This would completely change the structure of the document. Honestly, it confuses us, since it does not match the comments of the rest of the reviewers

  1. We believe that the structure of the document makes an analysis, comparison and description of the results in each study.The suggestion causes us confusion. What kind of figures are suggested?

We tried to summarize the information described in different studies

We do not believe that it is essential to include more figures. Again, this observation does not coincide with the comments of the rest of the reviewers.

  1. We value the comment and observation. Unfortunately, this suggestion confuses us

Within the structure of both documents (part 1 and 2) the possible mechanisms of action of each therapeutic effect are mentioned.

In section 3. "Chemical composition and mechanisms of pharmacological action of the Opuntia genus" they are also mentioned and related throughout the text.

In the "conclusions and perspectives" section, a summary of these mechanisms is made again.

We agree that there may be other undetermined mechanisms; however, trying to describe or represent other mechanisms of action would be creating a hypothesis and our objective is simply to analyze the studies that exist. We believe that the data described clearly suggest these mechanisms.

  1. We have considered the suggestion. A figure (Figure 2. Stages of skin healing) is included that describes the four phases of healing (hemostasis, inflammatory, proliferative, and remodeling).

In the rest of the text, the processes that exert an imbalance in healing are briefly described.

They are dividing into: chronic inflammatory state. keratinocyte involvement, and altered remodeling

Reviewer 2 Report

The original material presented here is of generally high quality - some of it immediately useful to my own work. I look forward to seeing the final version of this manuscript in print.

However, there are some problems and issues which must be addressed prior to publication, which go beyond the several examples of unclear expression noted below, viz:

Use of the phrase ‘synergistic activity’ should be further evidenced, or tempered in its presentation as more speculation than documented fact (see below for lines

There is far too much material here which has already been published elsewhere. This should be deleted from the manuscript, and (perhaps) a summary of relevant facts or frameworks provided in which this content can be cited as a reference.

As a general observation, there are far too many abbreviations, some appearing as acronyms, and very few of them in common circulation. This makes the paper challenging to read in places, and it is recommended to use only those abbreviations (such as chemical compounds), and not newly confected abbreviations, such as those by which the subspecies are distinguished.

There is too much detail in the review of some of the earlier studies, particularly across the seven pages of Table 1, the three pages of Table 2, the four pages of Table 3 and the two pages of Table 4. These should be edited for concision and reformatted into a table format, rather than the two columns of text, and presented so the column labels are visible on each page of the paper.

The narrative contains relict text, punctuation errors including using brackets instead of parentheses, and a recurrent formatting error of italicizing the abbreviation ‘spp.’ to mean multiple species (it should not be italicized, in line with the first part of the study (published separately).

23 Plants of the genus Opuntia spp

24 Specifically, Mexico

27 and other phenolic compounds

30 Part 1 collected information

34 potential, and its utility in

35 evidences of its beneficial properties

36 spp., not spp.

52 delete spp. (redundant)

57 ‘succulent plants’ (not a quotation)

58 very efficient in generating biomass

59 Noting that others have used the abbreviation ‘CLDs’ (not an acronym), it seems that ‘cladodes’ reads better.

61 [called cactus pear fruits (tunas) or prickly pear fruits (PPFs)]

Should be written as ‘- called cactus pear fruits, tunas, or prickly pear fruits (PPFs) - ’

65 Opuntia

67-68 ‘their domestication process in man-67 made environments has increased favoring the constant collection of CLD and PPFs’ is unclear – seemingly referring to the influence of human selection on the evolution of characters over time?

70 The genus Opuntia in Mexico

70-117: These two sections are reproduced wholesale from the ‘Part 1’ study already published. This content is not original to this publication, may thus be considered a form of ‘self-plagiarism’, and should be deleted from the manuscript.

118-122 acceptable duplication from the earlier study

123-136 this content has been reproduced wholesale from the ‘Part 1’ study, already published separately. This content is not original to this publication, may thus be considered a form of ‘self-plagiarism’, and should be deleted from the manuscript.

140 Here begins the original section of the paper.

142-158 citations required across this paragraph

162-170 citations required across this paragraph; brackets and parenthesis are again mixed up here

181-195 citations required across this paragraph

198 ‘neutrophils(Neut) and Macrop’ is unclear. Again, recommend you just use ‘neutrophils’ for the sake of clarity below.

201 Not clear what is meant by ‘have become an excellent source of procurement.’; spp., not ‘spp

223 Recommend deleting ‘In relation to the first evidence of the anti-ulcer effect,’. ‘Galati et al., (2001, 2002, 2003)’ is a different citation format than elsewhere in the paper; recommend you delete the years.

228 spp., not ‘spp

281-289,  292-302 citations required across these paragraphs; Quotation marks should not be used to present concepts; concepts should not be capitalized (viz line 308)

336-338 the word ‘and’ is missing from the sentence

347-359 citations required across this paragraph

375-378, 383-386, 408-412, 415-419, 444-449 etc. : citations required in these sentences

449 spp., not ‘spp

454 inconsistent citation of ‘Ginestra et al., (2009)’

449-501 sentence – a definition should be provided alongside this statement – what kind of ‘errors’ and how are they expressed ?

449-518 citations required across this paragraph

507 reduce, not decrease

523 what are the ‘numerous studies’? No citation is provided here – and only one follows in the text below.

597 e.g. and et al. should be italicised

643, 670 spp., not ‘spp

656-660 No citation is provided here – and it is not at all clear how such a ‘synergistic effect’ (a phrase which appears 4 times within this paper) has been explicitly investigated by any specific in vivo or in vitro study, including from references 60 and 80 (see below) nor in any of the 33 sources implicated in the overly broad citation of ‘[31-64]’ (should that read 31 and 64?).  Citations provided in support of these statements should specifically include evidence for synergistic activity.

From source 60 ‘The antiulcerogenic activity of OMFE might be due to a possible synergistic antioxidant and antihistaminic-like effects’ [emphasis added], which does not constitute a proof. As for source 80, it is written entirely in Korean, and although the abstract says that the study ‘verified the synergistic effects of combined treatment with EOFS and vitamin C,’ the evidence is not readily apparent.

Further evidence should be provided in support of this powerful idea – or else it should be presented as speculation – e.g. in line 657, where using the phrase ‘may be attributed’ instead of the more definite ‘are’ would be more objectively correct here.

More punctuation problems here with the brackets in this paragraph.

Author Response

Dear reviewer

The authors appreciate the comments and observations of the article

We have considered all suggestions and observations

Please check the attached file

Thanks for everything

Receive a cordial greeting

In summary:

  1. The wording of the phrases that mentioned "synergistic activity or effect" was modified. Effectively, Indeed, it is not correct to say so.

We better mention it as a possibility. Most of the studies that appear in the document mention a combined effect, therefore, we believe that it is not essential to add more references on this topic. Only the wording was changed.

  1. We appreciate the comment that "There is far too much material here which has already been published elsewhere". We believe that this was the main reason for making a compilation on these topics.

Continuing with the same format that was used for the publication of part 1, we have tried to assemble and summarize the information (Generalities, chemical composition and impact) of the Opuntia genus and of the scientific studies related to the topics described (All studies have been referenced according to the journal rules).

Other reviewers suggested that we consider more information on studies. Therefore, we adjust only a few paragraphs

  1. We have eliminated some abbreviations (which, indeed, can appear as acronyms). We are considering the most relevant and/or common abbreviations. The abbreviations CLDs and PPFs if they are used in other articles.In the case of the Opuntia subspecies, some of their abbreviations were only adjusted and considered to avoid extending the document size. We include in different places the full name of these abbreviations to make the document easier to read.
  2. Again, we appreciate the suggestion on the details, adjustment and revision of the studies.

We are continuing with the format established from part 1. We have tried to summarize some studies in the tables (including the full names of the subspecies).

The edition of the tables (such as part 1) will be adjusted by the editorial staff MDPI

  1. We have corrected punctuation errors, the use of brackets and/or parentheses, words and/or abbreviations written in italics (“spp”)

In the case of et al., (Effectively,, it should be in italics), unfortunately, the rules of the journal (in other articles already published it does not appear like this). However, we adjust it according to your observation

  1. We have attached all the references that were indicated to us. These references were found below those paragraphs. However, we consider the suggestion.
  2. We have included other tables and/or figures in the document.
  3. The scientific information on Opuntia is extensive and has been evidenced by different authors. Indeed, the sections (Introduction, Impact of the Opuntia genus in Mexico and other countries, Nutritional composition and mechanisms of pharmacological action of the Opuntia genus) presented in part 1 of these reviews is similar information to other authors. Since it corresponds to general aspects, chemical composition and impact of the Opuntia genus, which does not change or varies significantly from author to author (both in species, data, sources, values).

Some of the above information was used in this version 2. We believe that it cannot be considered self-plagiarism, since it is properly referenced and cited, according to the rules and authorization of the MDPI publisher.

In addition, there are more studies (evidence from articles are attached) where self-citation is used and information from the same authors is used.

We value feedback and to enrich this review, we have included more information and tables. In addition, some short paragraphs were adjusted. Our goal is to document and summarize the information on the Opuntia genus, which is very extensive.

Definition of plagiarism / self-plagiarism: Conscious act of appropriating ideas or texts, hiding the original source, by omitting to declare it or cite it in a different context or location that would make its identity recognized with the "new" text of the work

We welcome comments and feedback. We want the changes we have made to be sufficient for the document to be accepted

  1. a) Mark A. Moyad, M.D., M.P.H. Why a statin and/or another proven heart healthy agent should be utilized in the next major cancer chemoprevention trial: part I. Urologic Oncology: Seminars and Original Investigations 22 (2004) 466–471
  2. b) Mark A. Moyad, M.D., M.P.H. Why a statin and/or another proven heart healthy agent should be utilized in the next major cancer chemoprevention trial: part II. Urologic Oncology: Seminars and Original Investigations 22 (2004) 472–477
  3. c) Izquierdo-Vega JA, Morales-González JA, SánchezGutiérrez M, Betanzos-Cabrera G, Sosa-Delgado SM, Sumaya-Martínez MT, Morales-González Á, Paniagua-Pérez R, Madrigal-Bujaidar E, Madrigal-Santillán E. Evidence of Some Natural Products with Antigenotoxic Effects. Part 1: Fruits and Polysaccharides. 2017 Feb 2;9(2):102.
  4. d) López-Romero D, Izquierdo-Vega JA, Morales-González JA, Madrigal-Bujaidar E, Chamorro-Cevallos G, Sánchez-Gutiérrez M, Betanzos-Cabrera G, Alvarez-Gonzalez I, Morales-González Á, Madrigal-Santillán E. Evidence of Some Natural Products with Antigenotoxic Effects. Part 2: Plants, Vegetables, and Natural Resin. 2018 Dec 10;10(12):1954.
  5. f) Neumann DA, Camargo PR. Kinesiologic considerations for targeting activation of scapulo thoracic muscles - part 1: serratus anterior. Braz J PhysTher. 2019 Nov-Dec;23(6):459-466. AReview
  6. g) Camargo PR, Neumann DA. Kinesiologic considerations for targeting activation of scapulo thoracic muscles - part 2: trapezius. Braz J PhysTher. 2019 Nov-Dec;23(6):467-475. AReview
  7. h) Skelding A, Valverde A. Non-invasive blood pressure measurement in animals: Part 1 - Techniques for measurement and validation of non-invasive devices. Can Vet J. 2020 Apr;61(4):368-374.
  8. i) Skelding A, Valverde A. Non-invasive blood pressure measurement in animals: Part 2 - Evaluation of the performance of non-invasive devices. Can Vet J. 2020 May;61(5):481-498.

Line 23. The observation was considered and corrected.

Line 24. The observation was..”Specifically, Mexico”. We have confusion. What is the detail to adjust?

Line 27 The observation was..”and other phenolic compounds” We have confusion. What is the detail to adjust?

Line 30. The observation was..”Part 1 collected information” We have confusion. What is the detail to adjust?

Line 34 y 35. The observations were..”potential, and its utility in” and “evidences of its beneficial properties”. We have confusion. What is the detail to adjust?

Line 52. It was deleted “spp”

Line 57. The observation was considered and corrected.

Line 58. The observation was..” very efficient in generating biomass”. We have confusion. What is the detail to adjust?

Line 61. The observation was considered and corrected.

Line 67-68. The observation was considered and corrected. The wording was modified

Line 70. The observation was considered and corrected.

Line 201. The observation was considered and corrected. The wording was modified

Line 223. The observation was considered and corrected.

Line 281-289, 292-302. The observation was considered and corrected.

Line 336-338. The observation was considered and corrected.

Line 454. The observation was considered and corrected. Information is included

Line 449-501. The observation was considered and corrected. Information is included

Line 507. The observation was considered and corrected.

Line 523. The observation was considered and corrected. The wording was modified
